# PIXEL REWEIGHTED ADVERSARIAL TRAINING

## ABSTRACT

*Adversarial training* (AT) is a well-known defensive framework that trains a model with generated *adversarial examples* (AEs). AEs are crafted by intentionally adding perturbations to the natural images, aiming to mislead the model into making erroneous outputs. In existing AT methods, the magnitude of perturbations is usually constrained by a predefined perturbation budget, denoted as $\epsilon$, and *keeps the same* on each dimension of the image (i.e., each pixel within an image). However, in this paper, we discover that *not all pixels contribute equally* to the accuracy on AEs (i.e., robustness) and accuracy on natural images (i.e., accuracy). Motivated by this finding, we propose a new framework called ***P**ixel-reweighted **A**dve**R**sarial **T**raining (PART)*, to *partially* lower $\epsilon$ for pixels that rarely influence the model's outputs, which guides the model to focus more on regions where pixels are important for model's outputs. Specifically, we first use *class activation mapping* (CAM) methods to identify important pixel regions, then we keep the perturbation budget for these regions while lowering it for the remaining regions when generating AEs. In the end, we use these reweighted AEs to train a model. PART achieves a notable improvement in accuracy without compromising robustness on CIFAR-10, SVHN and Tiny-ImageNet and serves as a general framework, seamlessly integrating with a variety of AT, CAM and AE generation methods. More importantly, our work revisits the conventional AT framework and justifies the necessity to *allocate distinct weights to different pixel regions* during AT.

## 1 INTRODUCTION

Since the discovery of *adversarial examples* (AEs) by Szegedy et al. (2014), the security of deep learning models has become an area of growing concern, especially in critical applications such as autonomous driving. For instance, Kumar et al. (2020) show that by adding imperceptible adversarial noise, a well-trained model misclassifies a 'Stop' traffic sign as a 'Yield' traffic sign. To make sure the trained model is robust to AEs, *adversarial training* (AT) stands out as a representative defensive framework (Goodfellow et al., 2015; Madry et al., 2018), which trains a model with generated AEs. Normally, AEs are crafted by intentionally adding perturbations to the natural images, aiming to mislead the model into making erroneous outputs.

In existing AT methods, e.g., AT (Madry et al., 2018), TRADES (Zhang et al., 2019) and MART (Wang et al., 2020), the magnitude of perturbations (for generating AEs) is usually constrained by a predefined perturbation budget, denoted as $\epsilon$, and *keeps the same* on each dimension of the image (i.e., each pixel within an image) by assuming a $\ell_\infty$-norm constraint. We also analyze perturbations with $\ell_2$-norm constraint in Appendix A. Based on a $\ell_\infty$-norm constraint, one AE can be generated by solving the following constraint optimization problem:

$$\max_{\mathbf{\Delta}} \ell(f(\mathbf{x} + \mathbf{\Delta}), y), \text{ subject to } \|\mathbf{\Delta}\|_\infty \leq \epsilon, \qquad (1)$$

where $\ell$ is a loss function, $f$ is a model, $\mathbf{x} \in \mathbb{R}^d$ is a natural image, $y$ is the true label of $\mathbf{x}$, $\mathbf{\Delta} \in [-\epsilon, \epsilon]^d$ is the adversarial perturbation added to $\mathbf{x}$, $\|\cdot\|_\infty$ is the $\ell_\infty$-norm, $d$ is the data dimension, and $\epsilon$ is the maximum allowed perturbation budget. Let $\mathbf{\Delta}^*$ be the solution of the above optimization problem, then $\tilde{\mathbf{x}} = \mathbf{x} + \mathbf{\Delta}^*$ is the generated AE. Given that $\|\mathbf{\Delta}\|_\infty \leq \epsilon$, there is an implicit assumption in this AE generation process: all pixels have the *same* perturbation budget $\epsilon$. We argue that this assumption may overlook the fact that different pixel regions (e.g., textures, shapes, backgrounds, etc.) influence the model's outputs differently (Zhou et al., 2016; Selvaraju et al., 2017). For example, recent studies show that *convolutional neural networks* (CNNs) trained

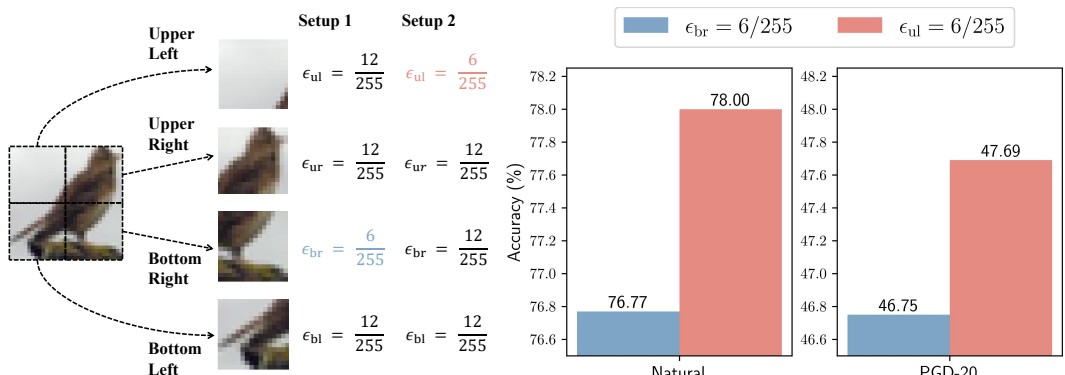

Figure 1: Fundamental discrepancies exist among different pixel regions. We segment each image into four equal-sized regions (i.e., ul, short for upper left; ur, short for upper right; br, short for bottom right; bl, short for bottom left) and adversarially train two ResNet-18 (He et al., 2016) on CIFAR-10 (Krizhevsky et al., 2009) using AT (Madry et al., 2018) with the same experiment settings except for the allocation of $\epsilon$. The robustness is evaluated by $\ell_\infty$-norm PGD-20 (Madry et al., 2018). With the same overall perturbation budgets (i.e., allocate one of the regions to $6/255$ and others to $12/255$), we find that both natural accuracy and adversarial robustness change significantly if the regional allocation on $\epsilon$ is different (e.g., by changing $\epsilon_{br} = 6/255$ to $\epsilon_{ul} = 6/255$, accuracy gains a 1.23% improvement and robustness gains a 0.94% improvement).

with ImageNet (Deng et al., 2009) are biased towards recognizing textures rather than shape in standard classification (Geirhos et al., 2019; Brendel & Bethge, 2019; Hermann & Lampinen, 2020).

Despite the insightful discussions made by the above studies, to the best of our knowledge, how the discrepancies of pixels would affect image classification in AT (i.e., robust classification) has not been well-investigated. Therefore, it is natural to raise the following question:

*Are all pixels equally important in robust classification?*

In this paper, we find that *not all pixels contribute equally* to the accuracy on AEs (i.e., robustness) and accuracy on natural images (i.e., accuracy). In Figure 1, we segment each image into four equal-sized regions and adversarially train two models with the same experiment settings except for the allocation of $\epsilon$. To clearly show the difference, we set $\epsilon = \{6/255, 12/255\}$. We keep the overall perturbation budget the same but allocate it differently among regions. From experimental results, we observe a significant improvement in both natural accuracy and adversarial robustness from the first setup to the second: natural accuracy increases from 76.77% to 78% and adversarial robustness increases from 46.75% to 47.69%. This means changing the perturbation budgets for different parts of an image has the potential to boost robustness and accuracy *at the same time*.

Motivated by this finding, we propose a new framework called ***Pixel-reweighted AdveRsarial Training (PART)***, to *partially* lower $\epsilon$ for pixels that rarely influence the model's outputs, which guides the model to focus more on regions where pixels are important for model's outputs.

To implement PART, we need to understand how pixels influence the model's output first. There are several well-known techniques to achieve this purpose, such as classifier-agnostic methods (e.g., LIME (Ribeiro et al., 2016)) and classifier-dependent methods (e.g., CAM, short for *class activation mapping* (Selvaraju et al., 2017; Fu et al., 2020; Jiang et al., 2021)). Given that classic AE generation processes are fundamentally classifier-dependent (Goodfellow et al., 2015; Madry et al., 2018), we choose to use CAM methods to identify the importance of pixels in terms of the influence on the model's outputs in PART. Then, we propose a ***Pixel-Reweighted AE Generation* (PRAG)** method. PRAG can keep the perturbation budget $\epsilon$ for important pixel regions while lowering the perturbation budget from $\epsilon$ to $\epsilon^{low}$ for the remaining regions when generating AEs. In the end, we can train a model with PRAG-generated AEs by using existing AT methods (e.g., AT (Madry et al., 2018), TRADES (Zhang et al., 2019), and MART (Wang et al., 2020)). We also analyze how perturbation budgets affect the generation of AEs given that features have unequal importance (see Theorem 1).

Through extensive evaluations on classic image datasets such as CIFAR-10 (Krizhevsky et al., 2009), SVHN Netzer et al. (2011) and Tiny-ImageNet (Wu, 2017), we demonstrate the effectiveness of PART (see Section 4). Despite a reduced overall perturbation budget, our approach not only significantly improves natural accuracy but also retains, and even marginally improves, robustness compared to existing defense methods. Moreover, PART is designed as a general framework that can be effortlessly incorporated with a variety of AT strategies (Madry et al., 2018; Zhang et al., 2019; Wang et al., 2020), CAM methods (Selvaraju et al., 2017; Fu et al., 2020; Jiang et al., 2021), and AE generation approaches (Madry et al., 2018; Gao et al., 2022).

In terms of PART's performance, the improvement in natural accuracy can be attributed to the sacrifice of the perturbation intensity for certain pixel regions, while the improvement in robustness seems counter-intuitive given the overall perturbation budget has been reduced. To deeply understand why PART can maintain or even improve the robustness, we take a close look at the robust feature representations. Previous studies (Tsipras et al., 2019; Ilyas et al., 2019) point out that the robust feature representations learnt by adversarially trained models align better with semantic information compared to standardly trained models. This means semantic information gains more robustness compared to non-semantic information, e.g., background information during the optimization procedure of AT. However, we find that without explicit guidance, it is hard for AT methods to fully align with semantic information. Based on this finding, we analyze the

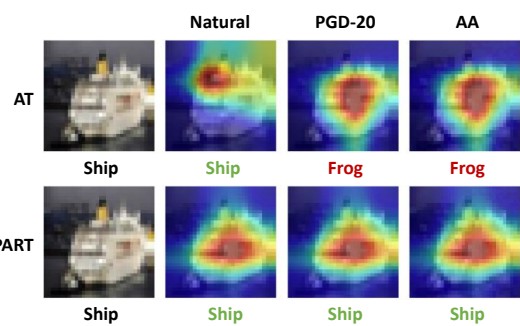

Figure 2: Vanilla AT vs. PART. The heatmaps are visualized by GradCAM (Selvaraju et al., 2017). The redder the color, the higher the contribution to classification result. PART aligns better with semantic information compared to vanilla AT. See Appendix B for more details.

vulnerability of neural networks in Appendix B. Our proposed method mitigates the problem by emphasizing the important pixel regions during training and thus provides external guidance to help models better extract features that are beneficial to robust classification (See Figure 2). We summarize the main contributions of our work as follows:

- We find that different pixel regions contribute differently to robustness and accuracy in robust classification. With the same total perturbation budget, allocating varying budgets to different pixel regions can improve robustness and accuracy simultaneously.

- We propose a new framework of AT, namely *Pixel-reweighted AdveRsarial Training (PART)* to guide the model focusing more on regions where pixels are important for model's output, leading to a better alignment with semantic information.

- We empirically show that, compared to the existing defenses, PART achieves a notable improvement in natural accuracy without compromising robustness on CIFAR-10, SVHN and Tiny-ImageNet against multiple attacks, including adaptive attacks.

## 1.1 RELATED WORK

**Reweighted adversarial training.** CAT (Cai et al., 2018) reweights adversarial data with different PGD iterations $K$. DAT (Wang et al., 2019) reweights the adversarial data with different convergence qualities. More recently, Ding et al. (2020) proposes to reweight adversarial data with instance-dependent perturbation bounds $\epsilon$ and Zhang et al. (2021) proposes a geometry-aware instance-reweighted AT framework (GAIRAT) that assigns different weights to adversarial loss based on the distance of data points from the class boundary. Our proposed method is fundamentally different from the existing methods. Existing reweighted AT methods primarily focus on instance-based reweighting, wherein each data instance is treated distinctly. PART, on the other hand, pioneers a pixel-based reweighting strategy, which allows for distinct treatment of pixel regions within each instance. Moreover, the design of PART is orthogonal to the state-of-the-art optimized AT methods (e.g., TRADES (Zhang et al., 2019) and MART (Wang et al., 2020)). This compatibility ensures that PART can be integrated into these established frameworks, thereby extending its utility.

**Class activation mapping.** CAM is a method for generating visual explanations of the decision made by a CNN for a given image. Given a class, CAM highlights the regions of the image that are most relevant to that class. Many CAM methods have been proposed to achieve the above purpose. For example, GradCAM (Selvaraju et al., 2017) improves upon vanilla CAM (Zhou et al., 2016) by using the gradient information flowing into the last convolutional layer of the CNN to assign importance values to each neuron, enabling the production of class-discriminative visualizations without the need for architectural changes or re-training. XGradCAM (Fu et al., 2020) introduces two axioms to improve the sensitivity and conservation of GradCAM. Specifically, it uses a modified gradient to better capture the importance of each feature map and a normalization term to preserve the spatial information of the feature maps. LayerCAM (Jiang et al., 2021) generates class activation maps not only from the final convolutional layer but also from shallow layers. This allows for both coarse spatial locations and fine-grained object details to be captured. We provide a more detailed related work in Appendix C.

## 2 PRELIMINARIES

**Adversarial training.** The basic idea behind AT (Madry et al., 2018) is to train a model $f$ with AEs generated from the original training data. The objective function of AT is defined as follows:

$$\min_{f \in F} \frac{1}{n} \sum_{i=1}^{n} \ell(f(\mathbf{x}_i + \mathbf{\Delta}_i^*), y_i), \tag{2}$$

where $\tilde{\mathbf{x}}_i = \mathbf{x}_i + \mathbf{\Delta}_i^*$ is the most adversarial variant of $\mathbf{x}_i$ within the $\epsilon$-ball centered at $\mathbf{x}_i$, $\mathbf{\Delta}_i^* \in [-\epsilon, \epsilon]^d$ is the optimized adversarial perturbation added to $\mathbf{x}_i$, $y_i$ is the true label of $\mathbf{x}_i$, $\ell$ is a loss function, and $F$ is the set of all possible neural network models.

The $\epsilon$-ball is defined as $B_\epsilon[\mathbf{x}] = \{\mathbf{x}' | \|\mathbf{x} - \mathbf{x}'\|_\infty \leq \epsilon\}$, where $\|\cdot\|_\infty$ is the $\ell_\infty$ norm. The most adversarial variant of $\mathbf{x}_i$ within the $\epsilon$-ball is commonly obtained by solving the constrained optimization problem in Eq. (1) using PGD (Madry et al., 2018):

$$\tilde{\mathbf{x}}_i^{(t+1)} = \tilde{\mathbf{x}}_i^{(t)} + \text{clip}(\tilde{\mathbf{x}}_i^{(t)} + \alpha \text{sign}(\nabla_{\tilde{\mathbf{x}}_i^{(t)}} \ell(f(\tilde{\mathbf{x}}_i^{(t)}), y_i)) - \mathbf{x}_i, -\epsilon, \epsilon), \tag{3}$$

where $\tilde{\mathbf{x}}_i^{(t)}$ is the AE at iteration $t$, $\alpha$ is the step size, $\text{sign}(\cdot)$ is the sign function, and $\text{clip}(\cdot, -\epsilon, \epsilon)$ is the clip function that projects the adversarial perturbation back into the $\epsilon$-ball, i.e., $\mathbf{\Delta}_i^* \in [-\epsilon, \epsilon]^d$.

**Class activation mapping.** In this paper, we mainly use GradCAM to identify the importance of the pixel regions because we find that the performance of PART with different CAM methods barely changes (see Section 4). Specifically, let $A_k \in R^{u \times v}$ of width $u$ and height $v$ for any class $c$ be the feature map obtained from the last convolutional layer of the CNN, and let $Y_c$ be the score for class $c$. GradCAM computes the gradient of $Y_c$ with respect to the feature map $A_k$:

$$\alpha_{c,k} = \frac{1}{Z} \sum_i \sum_j \frac{\partial Y_c}{\partial A_{k,ij}}, \tag{4}$$

where $Z$ is a normalization constant. GradCAM then produces the class activation map $L_c$ for class $c$ by computing the weighted combination of feature maps:

$$L_c = \text{ReLU}(\sum_k \alpha_{c,k} A_k). \tag{5}$$

## 3 PIXEL-REWEIGHTED ADVERSARIAL TRAINING

According the Figure 1, given the same overall perturbation budgets (i.e., allocate one of the regions to $6/255$ and others to $12/255$), we find that both natural accuracy and adversarial robustness change significantly if the regional allocation on $\epsilon$ is different. For example, by changing $\epsilon_{\text{br}} = 6/255$ to $\epsilon_{\text{ul}} = 6/255$, accuracy gains a 1.23% improvement and robustness gains a 0.94% improvement. This means changing the perturbation budgets for different parts of an image has the potential to boost robustness and accuracy *at the same time*. Thus, we consider a new framework, *Pixel-reweighted AdveRsarial Training* (PART), to train an adversarially robust model. In this section, we will introduce the learning objective, realization, and understanding of PART.

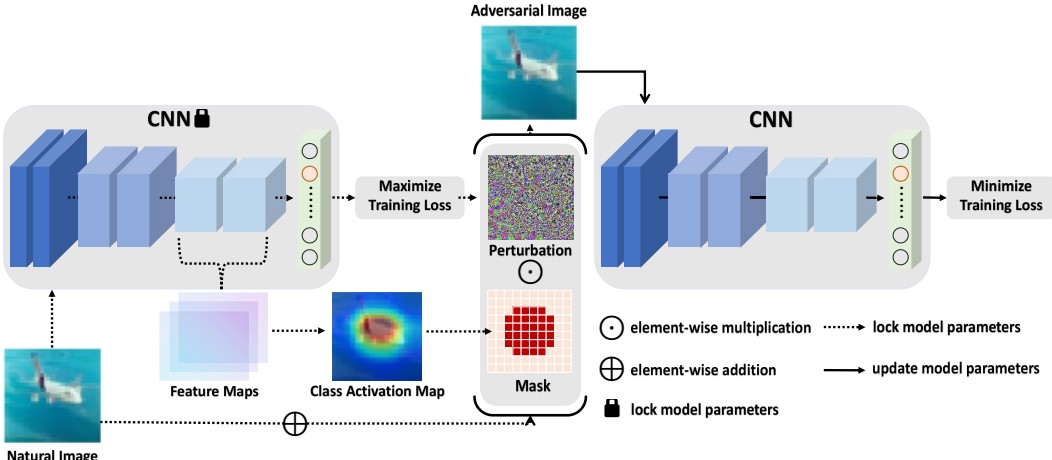

Figure 3: An overview of the training procedure for PART. Compared to AT, PART leverages the power of CAM methods to identify important pixel regions. Based on the class activation map, we element-wisely multiply a mask to the perturbation to keep the perturbation budget $\epsilon$ for important pixel regions while shrinking it to $\epsilon^{\text{low}}$ for their counterparts during the generation process of AEs.

## 3.1 LEARNING OBJECTIVE OF PART

Compared to the existing AT framework, PART will focus on generating AEs whose perturbation budget of each pixel may be different. Thus, we will first introduce the generation process of AEs within PART, and then conclude the learning objective of PART.

**AE generation process.** Compared to Eq. (1), the constraint optimization problem (for generating AEs in PART) will be:

$$\max_{\boldsymbol{\Delta}} \ell(f(\mathbf{x} + \boldsymbol{\Delta}), y), \text{ subject to } \|\boldsymbol{v}(\boldsymbol{\Delta}, \mathcal{I}^{\text{high}})\|_\infty \leq \epsilon, \|\boldsymbol{v}(\boldsymbol{\Delta}, \mathcal{I}^{\text{low}})\|_\infty \leq \epsilon^{\text{low}}, \quad (6)$$

where $\epsilon^{\text{low}} < \epsilon$, $\boldsymbol{\Delta} = [\delta_1, \ldots, \delta_d]$, $\mathcal{I}^{\text{high}}$ collect indexes of important pixels, $\mathcal{I}^{\text{low}} = [d]/\mathcal{I}^{\text{high}}$, and $\boldsymbol{v}$ is a function to transform a set (e.g., a set consisting of important pixels in $\boldsymbol{\Delta}$: $\{\delta_i\}_{i \in \mathcal{I}^{\text{high}}}$) to a vector. Then, $\boldsymbol{\Delta}^{\text{high}}$ consists of $\{\delta_i\}_{i \in \mathcal{I}^{\text{high}}}$, and $\boldsymbol{\Delta}^{\text{low}}$ consists of $\{\delta_i\}_{i \in \mathcal{I}^{\text{low}}}$. $\boldsymbol{\Delta}^{\text{high}} \in [-\epsilon, \epsilon]^{d^{\text{high}}}$ is the adversarial perturbation added to important pixel regions with dimension $d^{\text{high}}$, $\boldsymbol{\Delta}^{\text{low}} \in [-\epsilon^{\text{low}}, \epsilon^{\text{low}}]^{d^{\text{low}}}$ is the adversarial perturbation added to the remaining regions with dimension $d^{\text{low}}$, where $d^{\text{high}} = |\mathcal{I}^{\text{high}}|$ and $d^{\text{low}} = |\mathcal{I}^{\text{low}}|$. A higher value of $d^{\text{high}}$ means that more pixels are regarded as important ones. A detailed description of notations can be found in Appendix D.

**Learning objective.** Given a training set $\{\mathbf{x}_i, y_i\}_{i=1}^n$, a loss function $\ell$, a function space $F$, and the largest perturbation budget $\epsilon$, the PART-based algorithms should have the same learning objective:

$$\min_{f \in F} \frac{1}{n} \sum_{i=1}^n \ell(f(\mathbf{x}_i + \boldsymbol{\Delta}_i^*), y_i), \quad (7)$$

$$\boldsymbol{\Delta}_i^* = \arg\max_{\boldsymbol{\Delta}} \ell(f(\mathbf{x}_i + \boldsymbol{\Delta}), y_i), \text{ subject to } \|\boldsymbol{v}(\boldsymbol{\Delta}, \mathcal{I}^{\text{high}})\|_\infty \leq \epsilon, \|\boldsymbol{v}(\boldsymbol{\Delta}, \mathcal{I}^{\text{low}})\|_\infty \leq \epsilon^{\text{low}}. \quad (8)$$

Compared to other frameworks, the learning objective of PART is clearly different from theirs in terms of the AE generation process. In the following subsection, we will introduce how to achieve the above learning objective via an empirical method.

## 3.2 REALIZATION OF PART

**Pixel-reweighted AE generation (PRAG).** The constraint optimization problem Eq. (6) implies that the overall perturbation $\boldsymbol{\Delta}$ consists of two parts: perturbation added to important pixel regions, i.e., $\boldsymbol{\Delta}^{\text{high}}$ and perturbation added to their counterparts, i.e., $\boldsymbol{\Delta}^{\text{low}}$.

To generate AEs with appropriate $\boldsymbol{\Delta}^{\text{high}}$ and $\boldsymbol{\Delta}^{\text{low}}$, we propose a method called *Pixel-Reweighted AE Generation* (PRAG). PRAG employs CAM methods to differentiate between important pixel

regions and their counterparts. Take GradCAM as an example: once we compute the class activation map $L_c$ from Eq. (5), PRAG first resizes $L_c$ to $L'_c$ to match the dimensions $d$ of a natural image $\mathbf{x} = [x_1, ..., x_d]$, i.e., $L'_c \in \mathbb{R}^d$. Then it scales $L'_c$ to $\tilde{L}_c$ to make sure the pixel regions highlighted by GradCAM have a weight value $\omega > 1$. Let $\tilde{L}_c = [\omega_1, ..., \omega_d]$ and $\mathbf{\Delta} = [\delta_1, ..., \delta_d]$ consists of $\mathbf{\Delta}^{\text{high}}$ and $\mathbf{\Delta}^{\text{low}}$. Then, for any $i \in [d]$, we define $\delta_i \in \mathbf{\Delta}^{\text{high}}$ if $\omega_i > 1$ and $\delta_i \in \mathbf{\Delta}^{\text{low}}$ otherwise, subject to $\|\boldsymbol{v}(\mathbf{\Delta}, \mathcal{I}^{\text{high}})\|_\infty \leq \epsilon$ and $\|\boldsymbol{v}(\mathbf{\Delta}, \mathcal{I}^{\text{low}})\|_\infty \leq \epsilon^{\text{low}}$. Technically, this is equivalent to element-wisely multiply a mask $\boldsymbol{m} = [m_1, ..., m_d]$ to a $\mathbf{\Delta}$ constraint by $\|\mathbf{\Delta}\|_\infty \leq \epsilon$, where each element of $\boldsymbol{m}$ is defined as:

$$m_i = \begin{cases} 1 & \text{if } \omega_i > 1 \\ \epsilon^{\text{low}}/\epsilon & \text{otherwise} \end{cases}. \tag{9}$$

Let $\mathbf{\Delta}^*$ be the optimal solution of $\mathbf{\Delta}$, then $\tilde{\mathbf{x}} = \mathbf{x} + \mathbf{\Delta}^*$ is the AE generated by PRAG, which can be obtained by solving Eq. (8) using an adapted version of Eq. (3):

$$\tilde{\mathbf{x}}_i^{(t+1)} = \tilde{\mathbf{x}}_i^{(t)} + \boldsymbol{m} \odot \text{clip}(\tilde{\mathbf{x}}_i^{(t)} + \alpha \text{sign}(\nabla_{\tilde{\mathbf{x}}_i^{(t)}} \ell(f(\tilde{\mathbf{x}}_i^{(t)}), y_i)) - \mathbf{x}_i, -\epsilon, \epsilon), \tag{10}$$

where $\odot$ is the Hadamard product. By doing so, we element-wisely multiply a mask $\boldsymbol{m}$ to the perturbation to keep the perturbation budget $\epsilon$ for important pixel regions while shrinking it to $\epsilon^{\text{low}}$ for their counterparts. We provide a visual illustration of the training procedure for PART in Figure 3 and a detailed algorithmic description in Appendix E.

**How to select $\epsilon^{\text{low}}$.** Given that the value of $\epsilon^{\text{low}}$ is designed to be a small number (e.g., $6/255$) and the computational cost of AT is expensive, we do not apply any algorithms to search for an optimal $\epsilon^{\text{low}}$ to avoid introducing extra training time to our framework. Instead, we directly set $\epsilon^{\text{low}} = \epsilon - 1/255$ by default. Without losing generality, we thoroughly investigate the impact of different values of $\epsilon^{\text{low}}$ on the robustness and accuracy of our method (see Section 4). Designing an efficient searching algorithm for $\epsilon$ remains an open question, and we leave it as our future work.

**Burn-in period.** To improve the effectiveness of PART, we integrate a *burn-in period* into our training process. Specifically, we use AT as a warm-up at the early stage of training. Then, we incorporate PRAG into PART for further training. This is because the classifier is not properly learned initially, and thus may badly identify pixel regions that are important to the model's output.

**Integration with other methods.** The innovation on the AE generation allows PART to be orthogonal to the commonly used AT methods (e.g., AT (Madry et al., 2018), TRADES (Zhang et al., 2019) and MART (Wang et al., 2020)), and thus PART can be easily integrated into these established methods. Moreover, the constraint optimization problem in Eq. (8) is general and can be addressed using various existing algorithms, such as PGD (Madry et al., 2018) and MMA (Gao et al., 2022). Besides, many CAM methods can be used as alternatives to GradCAM, such as XGradCAM (Fu et al., 2020) and LayerCAM (Jiang et al., 2021). Therefore, the compatibility of PART allows itself to serve as a general framework.

## 3.3 How Perturbation Budgets Affect the Generation of AEs

We study a toy setting to shed some light on how pixels with different levels of importance would affect the generated AEs. Consider a 2D data point $\mathbf{x} = [x_1, x_2]^T$ with label $y$ and an adversarial perturbation $\mathbf{\Delta} = [\delta_1, \delta_2]^T$ that is added to $\mathbf{x}$ with $\delta_1 \in [-\epsilon_1, \epsilon_1]$ and $\delta_2 \in [-\epsilon_2, \epsilon_2]$, where $\epsilon_1$ and $\epsilon_2$ are maximum allowed perturbation budgets for $\delta_1$ and $\delta_2$, respectively. Let $\ell$ be a differentiable loss function and $f$ be the model, The constraint optimization problem (used to generate AEs) can be formulated as follows:

$$\max_{\mathbf{\Delta}=[\delta_1, \delta_2]^T} \ell(f(\mathbf{x} + \mathbf{\Delta}), y), \text{ subject to } -\epsilon_1 \leq \delta_1 \leq \epsilon_1, \ -\epsilon_2 \leq \delta_2 \leq \epsilon_2. \tag{11}$$

Then, based on the *Karush–Kuhn–Tucker* (KKT) conditions (Avriel, 2003) for constraint optimization problems, we can analyze the solutions to the above problem as follows.

**Lemma 1.** *Let $\delta_1^*$ and $\delta_2^*$ be the optimal solutions of Eq.* (11)*. The generated AEs can be categorized into three cases: (i) The expressions of $\delta_1^*$ and $\delta_2^*$ do not contain $\epsilon_1$ and $\epsilon_2$. (ii) $\delta_1^* = \pm\epsilon_1$ and $\delta_2^* = \pm\epsilon_2$. (iii) $\delta_1^* = \pm\epsilon_1$ and $\delta_2^*$ is influenced by $\epsilon_1$, or vise versa.*

From Lemma 1, we know that the generated AEs must be within these cases, as KKT provides necessary conditions that $\delta_1^*$ and $\delta_2^*$ must satisfy. Nevertheless, for different models, the solutions

are different. Here we focus on the impact on linear models. Specifically, we consider a linear model $f(\mathbf{x}) = \omega_1 x_1 + \omega_2 x_2 + b$ for this problem, where $\omega_1$ and $\omega_2$ are the weights for pixels $x_1$ and $x_2$ respectively. It is clear that $x_1$ will significantly influence $f(\mathbf{x})$ more compared to $x_2$ if $w_1$ is larger than $w_2$. For simplicity, we use a square loss, which can be expressed as $\ell(f(\mathbf{x}), y) = (y - f(\mathbf{x}))^2$. Then, we solve Eq. (11) by the Lagrange multiplier method and show the results in Theorem 1.

**Theorem 1.** *Consider a linear model $f(\mathbf{x}) = \omega_1 x_1 + \omega_2 x_2 + b$ and a square loss $\ell(f(\mathbf{x}), y) = (y - f(\mathbf{x}))^2$. Let $\delta_1^*$ and $\delta_2^*$ be the optimal solutions of Eq. (11). For case (iii) in Lemma 1, we have:*

$$\delta_2^* = \frac{y - f(\mathbf{x}) - \omega_1 \epsilon_1}{\omega_2}, \text{ subject to } \delta_1^* = \epsilon_1, \tag{12}$$

$$\delta_2^* = \frac{y - f(\mathbf{x}) + \omega_1 \epsilon_1}{\omega_2}, \text{ subject to } \delta_1^* = -\epsilon_1, \tag{13}$$

$$\delta_1^* = \frac{y - f(\mathbf{x}) - \omega_2 \epsilon_2}{\omega_1}, \text{ subject to } \delta_2^* = \epsilon_2, \tag{14}$$

$$\delta_1^* = \frac{y - f(\mathbf{x}) + \omega_2 \epsilon_2}{\omega_1}, \text{ subject to } \delta_2^* = -\epsilon_2. \tag{15}$$

We provide a more detailed analysis and the proof of Lemma 1 and Theorem 1 in Appendix F. From Theorem 1, the main takeaway is straightforward: If two pixels have different influences on the model's predictions, it will affect the generation process of AEs, leading to different solutions of the optimal $\delta^*$. Thus, it probably influences the performance of AT.

**Remark.** Note that, we do not cover how different levels of pixel importance would affect the performance of AT. This is because, during AT, the generated AEs are highly correlated, making the training process quite complicated to analyze in theory. According to recent developments regarding learning with dependent data (Dagan et al., 2019), we can only expect generalization when weak dependence exists in training data. However, after the first training epoch in AT, the model already depends on all training data, meaning that the generated AEs in the following epochs are probably highly dependent on each other. Thus, we leave this as our future work.

## 4 EXPERIMENTS

**Dataset.** We evaluate the effectiveness of PART on three benchmark datasets, i.e., CIFAR-10 (Krizhevsky et al., 2009), SVHN (Netzer et al., 2011) and Tiny-ImageNet (Wu, 2017). CIFAR-10 comprises 50,000 training and 10,000 test images, distributed across 10 classes, with a resolution of $32 \times 32$. SVHN has 10 classes but consists of 73,257 training and 26,032 test images, maintaining the same $32 \times 32$ resolution. Tiny-ImageNet extends the complexity by offering 200 classes with a higher resolution of $64 \times 64$, containing 100,000 training, 10,000 validation, and 10,000 test images. For the target model, following the idea in Zhou et al. (2023), we use ResNet (He et al., 2016) for CIFAR-10 and SVHN, and WideResNet (Zagoruyko & Komodakis, 2016) for Tiny-ImageNet.

**Attack settings.** We mainly use three types of adversarial attacks to evaluate the performances of defenses. They are $\ell_\infty$-norm PGD (Madry et al., 2018), $\ell_\infty$-norm MMA (Gao et al., 2022) and $\ell_\infty$-norm AA (Croce & Hein, 2020a). Among them, AA is a combination of three non-target white-box attacks (Croce & Hein, 2020b) and one targeted black-box attack (Andriushchenko et al., 2020), which makes AA a gold standard for evaluating adversarial robustness up to this point. Recently proposed MMA (Gao et al., 2022) can achieve comparable performance compared to AA but is much more time efficient. The iteration number for PGD is set to 20 (Zhou et al., 2023), and the target selection number for MMA is set to 3 (Gao et al., 2022), respectively. For all attacks, we set the maximuim allowed perturbation budget $\epsilon$ to $8/255$.

**Defense settings.** We use three representative AT methods as the baselines: AT (Madry et al., 2018) and two optimized AT methods TRADES (Zhang et al., 2019) and MART (Wang et al., 2020). We set $\lambda = 6$ for both TRADES and MART. For all baseline methods, we use the $\ell_\infty$-norm non-targeted PGD-10 with random start to craft AEs in the training stage. We set $\epsilon = 8/255$ for all methods, and $\epsilon^{\text{low}} = 7/255$ for PART. More details can be found in Appendix G.

Table 1: Robustness (%) and accuracy (%) of defense methods on *CIFAR-10*, *SVHN* and *Tiny-ImageNet*. We report the averaged results and standard deviations of three runs. We show the most successful defense in **bold**.

| | | ResNet-18 | | | |
|---|---|---|---|---|---|
| Dataset | Method | Natural | PGD-20 | MMA | AA |
| CIFAR-10 | AT | $82.58 \pm 0.14$ | $\mathbf{43.69 \pm 0.28}$ | $41.80 \pm 0.10$ | $41.63 \pm 0.22$ |
| | PART | $\mathbf{83.42 \pm 0.26}$ | $43.65 \pm 0.16$ | $\mathbf{41.98 \pm 0.03}$ | $\mathbf{41.74 \pm 0.04}$ |
| | TRADES | $78.16 \pm 0.15$ | $48.28 \pm 0.05$ | $45.00 \pm 0.08$ | $45.05 \pm 0.12$ |
| | PART-T | $\mathbf{79.36 \pm 0.31}$ | $\mathbf{48.90 \pm 0.14}$ | $\mathbf{45.90 \pm 0.07}$ | $\mathbf{45.97 \pm 0.06}$ |
| | MART | $76.82 \pm 0.28$ | $49.86 \pm 0.32$ | $45.42 \pm 0.04$ | $45.10 \pm 0.06$ |
| | PART-M | $\mathbf{78.67 \pm 0.10}$ | $\mathbf{50.26 \pm 0.17}$ | $\mathbf{45.53 \pm 0.05}$ | $\mathbf{45.19 \pm 0.04}$ |
| SVHN | AT | $91.06 \pm 0.24$ | $49.83 \pm 0.13$ | $47.68 \pm 0.06$ | $45.48 \pm 0.05$ |
| | PART | $\mathbf{93.14 \pm 0.05}$ | $\mathbf{50.34 \pm 0.14}$ | $\mathbf{48.08 \pm 0.09}$ | $\mathbf{45.67 \pm 0.13}$ |
| | TRADES | $88.91 \pm 0.28$ | $\mathbf{58.74 \pm 0.53}$ | $53.29 \pm 0.56$ | $52.21 \pm 0.47$ |
| | PART-T | $\mathbf{91.55 \pm 0.21}$ | $58.64 \pm 0.26$ | $\mathbf{53.84 \pm 0.16}$ | $\mathbf{52.31 \pm 0.67}$ |
| | MART | $89.76 \pm 0.08$ | $58.52 \pm 0.53$ | $52.42 \pm 0.34$ | $49.10 \pm 0.23$ |
| | PART-M | $\mathbf{91.42 \pm 0.36}$ | $\mathbf{58.85 \pm 0.29}$ | $\mathbf{52.45 \pm 0.03}$ | $\mathbf{49.92 \pm 0.10}$ |
| | | WideResNet-34-10 | | | |
| Dataset | Method | Natural | PGD-20 | MMA | AA |
| Tiny-ImageNet | AT | $43.51 \pm 0.13$ | $11.70 \pm 0.08$ | $10.66 \pm 0.11$ | $10.53 \pm 0.14$ |
| | PART | $\mathbf{44.87 \pm 0.21}$ | $\mathbf{11.93 \pm 0.16}$ | $\mathbf{10.96 \pm 0.12}$ | $\mathbf{10.76 \pm 0.06}$ |
| | TRADES | $43.05 \pm 0.15$ | $13.86 \pm 0.10$ | $12.62 \pm 0.16$ | $12.55 \pm 0.09$ |
| | PART-T | $\mathbf{44.31 \pm 0.12}$ | $\mathbf{14.08 \pm 0.22}$ | $\mathbf{13.01 \pm 0.09}$ | $\mathbf{12.84 \pm 0.14}$ |
| | MART | $42.68 \pm 0.22$ | $14.77 \pm 0.18$ | $13.58 \pm 0.13$ | $13.42 \pm 0.16$ |
| | PART-M | $\mathbf{43.75 \pm 0.24}$ | $\mathbf{14.93 \pm 0.15}$ | $\mathbf{13.76 \pm 0.06}$ | $\mathbf{13.68 \pm 0.13}$ |

Table 2: Robustness (%) of defense methods against adaptive PGD on CIFAR-10. We report the averaged results and standard deviations of three runs. We show the most successful defense in **bold**.

| | | ResNet-18 | | | | |
|---|---|---|---|---|---|---|
| Dataset | Method | PGD-20 | PGD-40 | PGD-60 | PGD-80 | PGD-100 |
| CIFAR-10 | AT | $37.67 \pm 0.05$ | $36.98 \pm 0.03$ | $36.86 \pm 0.07$ | $36.81 \pm 0.04$ | $36.72 \pm 0.04$ |
| | PART | $\mathbf{37.73 \pm 0.11}$ | $\mathbf{37.07 \pm 0.08}$ | $\mathbf{36.89 \pm 0.12}$ | $\mathbf{36.84 \pm 0.10}$ | $\mathbf{36.84 \pm 0.07}$ |
| | TRADES | $43.42 \pm 0.13$ | $43.22 \pm 0.11$ | $43.19 \pm 0.12$ | $43.10 \pm 0.08$ | $43.08 \pm 0.06$ |
| | PART-T | $\mathbf{43.98 \pm 0.15}$ | $\mathbf{43.75 \pm 0.09}$ | $\mathbf{43.73 \pm 0.06}$ | $\mathbf{43.68 \pm 0.10}$ | $\mathbf{43.61 \pm 0.03}$ |
| | MART | $44.60 \pm 0.09$ | $44.19 \pm 0.14$ | $44.05 \pm 0.13$ | $43.98 \pm 0.05$ | $43.96 \pm 0.08$ |
| | PART-M | $\mathbf{44.96 \pm 0.21}$ | $\mathbf{44.51 \pm 0.17}$ | $\mathbf{44.41 \pm 0.12}$ | $\mathbf{44.37 \pm 0.06}$ | $\mathbf{44.35 \pm 0.09}$ |

**Defending against general attacks.** From Table 1, the results show that our method can notably improve the natural accuracy with little to no degradation in adversarial robustness compared to AT. Despite a marginal reduction in robustness by 0.04% on PGD-20, PART gains more on natural accuracy (e.g., 2.08% on SVHN and 1.36% on Tiny-ImageNet). In most cases, PART can improve natural accuracy and robustness simultaneously. To avoid the bias caused by different AT methods, we apply the optimized AT methods TRADES and MART to our method (i.e., PART-T and PART-M). Compared to TRADES and MART, our method can still boost natural accuracy (e.g., 1.20% on CIFAR-10, 2.64% on SVHN and 1.26% on Tiny-ImageNet for PART-T, and 1.85% on CIFAR-10, 1.66% on SVHN and 1.07% on Tiny-ImageNet) with at most a 0.10% drop in robustness, and thus our method can achieve a better robustness-accuracy trade-off. Besides, we consider the five behaviours listed in Athalye et al. (2018) to identify the obfuscated gradients and show that our method does not cause obfuscated gradients (see Appendix H).

**Defending against adaptive attacks.** Beyond general adversarial attacks, a more destructive adaptive attack strategy has been proposed to evaluate the robustness of the defense methods. This strategy assumes attacks have all the knowledge about the proposed method, e.g., model architectures, model parameters, and how AEs are generated in PART. As a result, attackers can design a specific attack to break PART. Given the details of PRAG, we design an adaptive attack that aims to misguide the model to focus on pixel regions that have little contribution to the correct classification results, and thus break the defense. We provide relevant explanations in Appendix B. Technically,

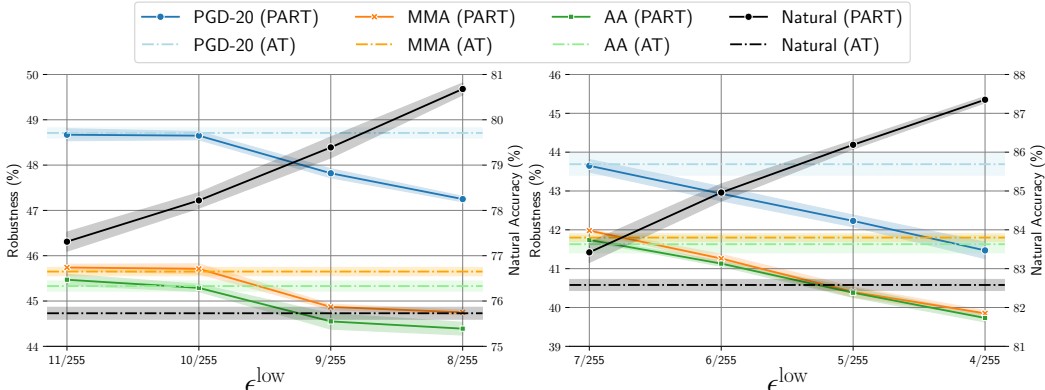

Figure 4: Impact of $\epsilon^{\text{low}}$ on robustness and accuracy of PART. **Left:** $\epsilon = 12/255$ and $\epsilon^{\text{low}} \in \{11/255, 10/255, 9/255, 8/255\}$. **Right:** $\epsilon = 8/255$, and $\epsilon^{\text{low}} \in \{7/255, 6/255, 5/255, 4/255\}$. Solid lines represent the performance of PART, and dashed lines represent the performance of AT. We report the averaged results and standard deviations (i.e., shaded areas) of three runs.

this is equivalent to breaking what a robust model currently focuses on. Specifically, we use PRAG with PGD to craft AEs, with an increased $\epsilon^{\text{low}}$ of $8/255$ and $\epsilon$ of $12/255$. As shown in Table 2, despite an overall decrease in robustness, our defense presents a better resilience against adaptive attacks compared to other baseline methods. More experiments can be found in Appendix I and J.

**Hyperparameter analysis.** We thoroughly investigated the impact of the hyperparameter $\epsilon^{\text{low}}$ on the effectiveness of our method. We consider two sets of experiments. In the first set, we set $\epsilon = 12/255$ and $\epsilon^{\text{low}} \in \{11/255, 10/255, 9/255, 8/255\}$. In the second set, we set $\epsilon = 8/255$, and $\epsilon^{\text{low}} \in \{7/255, 6/255, 5/255, 4/255\}$. As shown in Figure 4, with the decrease of $\epsilon^{\text{low}}$, the robustness of the model drops correspondingly. However, PART gains more natural accuracy at the same time, and thus achieves a better robustness-accuracy trade-off. In addition, we find that with a relatively large $\epsilon$, moderately decrease $\epsilon^{\text{low}}$ barely changes the robustness. For example, our method achieves a notable improvement in natural accuracy without compromising robustness when $\epsilon = 12/255$ and $\epsilon^{\text{low}} \in \{11/255, 10/255\}$ compared to AT.

**Integration with other CAM methods.** To avoid potential bias caused by different CAM methods, we conduct experiments to compare the performance of PART with different CAM methods such as GradCAM (Selvaraju et al., 2017), XGradCAM (Fu et al., 2020) and LayerCAM (Jiang et al., 2021). We find that these state-of-the-art CAM methods have approximately identical performance (see Appendix K). Thus, we argue that the performance of PART is barely affected by the choice of benchmark CAM methods.

**Integration with other AE generation methods.** In addition, we evaluate the effectiveness of our method by incorporating PRAG into a more destructive attack, i.e., MMA (Gao et al., 2022) to generate AEs. With MMA, the performance of PART can be further boosted. (see Appendix L).

**Training speed and memory consumption of PART.** To avoid introducing unaffordable extra cost by CAM methods, we update the mask $m$ for every 10 epochs. We compare the computational time and the memory consumption of our method to different baseline methods (see Appendix M).

## 5 CONCLUSION

We find that different pixel regions contribute unequally to robustness and accuracy. Motivated by this finding, we propose a new framework called *Pixel-reweighted AdveRsarial Training (PART)*. PART partially reduces the perturbation budget for pixel regions that rarely influence the classification results, which guides the classifier to focus more on the essential part of images, leading to a better alignment with semantic information. In general, we hope this simple yet effective framework could open up a new perspective in AT and lay the groundwork for advanced defenses that account for the discrepancies across pixel regions.

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

## A    PERTURBATIONS WITH $\ell_2$-NORM CONSTRAINT

When discussing perturbations with $\ell_2$-norm constraint, it's not accurate to assume each pixel has the same perturbation budget $\epsilon$. This is because compared to a $\ell_\infty$-norm constraint, the entire perturbation $\boldsymbol{\Delta}$ is subject to a global bound, rather than each dimension having an identical perturbation budget. Let the dimension of a natural image $\mathbf{x}$ be $d$. For a perturbation $\boldsymbol{\Delta} = [\delta_1, ..., \delta_d]$, we have:

$$\|\boldsymbol{\Delta}\|_2 = \sqrt{\delta_1^2 + \delta_2^2 + ... + \delta_d^2} \le \epsilon, \tag{16}$$

where $\epsilon$ is the maximum allowed perturbation budget. By Eq. (16), $\delta_i$ is not necessarily less than or equal to $\epsilon$, e.g., certain elements might undergo minimal perturbations approaching 0, while others might be more significantly perturbed, as long as the entire vector's $\ell_2$-norm remains under $\epsilon$.

Thus, in this paper, the assumption that all pixels have the *same* perturbation budget $\epsilon$ is discussed by assuming the perturbations are bounded by $\ell_\infty$-norm constraint, i.e., $\|\boldsymbol{\Delta}\|_\infty \le \epsilon$.

## B    ANALYSIS OF VULNERABILITY OF NEURAL NETWORKS AND BEYOND

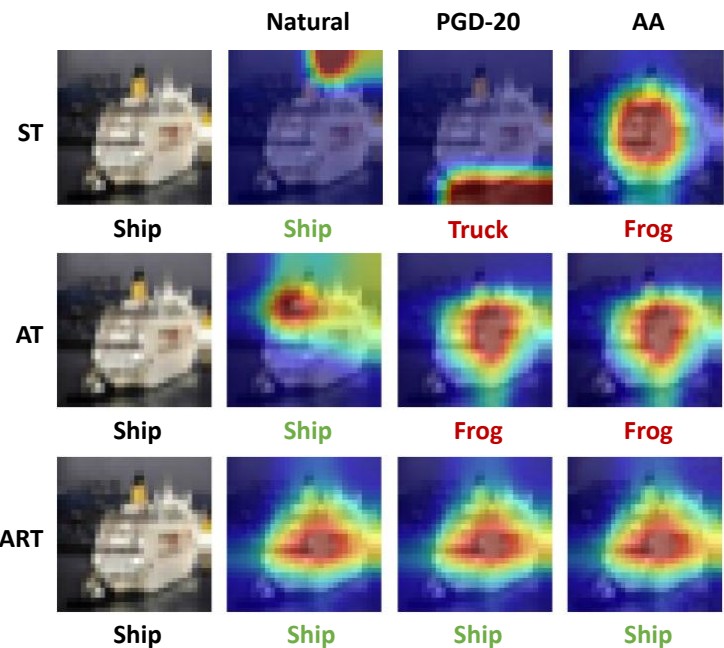

Figure 5: Standardly trained (ST) model vs. adversarially trained model by AT (Madry et al., 2018) vs. adversarially trained model by PART. The heatmaps are visualized by GradCAM (Selvaraju et al., 2017).The redder the color, the higher the contribution to classification result. PART aligns better with semantic infomration compared to AT and ST. The target model is ResNet-18 and the dataset is CIFAR-10.

We want to provide intuitions on the vulnerabilities of deep neural networks through the lens of how adversarial attacks will affect the weights of pixels. To achieve this, we compare the *standardly trained* (ST) model and adversarially trained models by visualizing the weight values of the corresponding feature maps, i.e., the class activation map via GradCAM against PGD-20 and AA.

In Figure 5, we define the highlighted pixel regions in the *second column*, i.e., labelled as 'Natural', as the *important pixel regions* to the correct prediction of each model, and its counterpart of the image as the *unimportant pixel regions* to the correct prediction. To produce correct prediction results, the model relies heavily on the important pixel regions as the weights of these regions are much larger than unimportant pixel regions. A successful attack, however, will misguide the model to pay more attention to unimportant pixel regions (e.g., PGD-20 misguides the ST model to focus

on the bottom part of the ship, which is far from its important pixel regions), and thus produces erroneous results because these pixels are hardly contributing to the correct prediction.

Besides, we find that the ST model and AT model have completely different focus. For example, a ST model focuses more on the background of the image while an AT model focuses more on the object of the image. This finding matches the statement of Tsipras et al. (2019). It is a benefit of AT as we hope deep neural networks can mimic the way humans recognize objects (e.g., recognize an object by its semantic meaning). During the optimization procedure of AT, the model is forced to learn the underlying distributions of AEs and extract robust features to defend against adversarial attacks. Thus, semantic information gains more robustness compared to the background information, as they are naturally highlighted during the optimization procedure of AT.

However, without explicit guidance, we find that it is hard for the AT model to fully align with semantic information. Although the AT model is much more aligned with the semantic features of the ship compared to the ST model, adversarial attacks can still misguide the AT model to pay more attention to other unimportant pixel regions (see Figure 5). This property provides intuition on how to better defend against these attacks. That is, if we can provide external guidance to train a model that can stably align with semantic information when facing adversarial attacks, the robustness should be further improved. PART can be regarded as one approach to help models align better with semantic information. We provide more evidence in Appendix N. We believe there exist some other methods to achieve the same purpose and we hope our work can promote the development of object-aligned frameworks.

## C  DETAILED RELATED WORK

**Adversarial training.** To combat the threat of adversarial attacks, a myriad of defense mechanisms have emerged, such as perturbation detection (Ma et al., 2018; Xu et al., 2018; Gao et al., 2021), adversarial purification (Shi et al., 2021; Yoon et al., 2021; Nie et al., 2022) and adversarial training (AT) (Madry et al., 2018; Zhang et al., 2019; Wang et al., 2020). Among these, AT stands out as a representative strategy (Goodfellow et al., 2015; Madry et al., 2018), which directly generates and incorporates AEs during the training process, forcing the model to learn the underlying distributions of AEs. Besides vanilla AT (Madry et al., 2018), many alternatives have been proposed. For example, from the perspective of improving objective functions, Zhang et al. (2019) proposes to optimize a surrogate loss function, which is derived based on a theoretical upper bound and a lower bound. Wang et al. (2020) investigates the unique impact of misclassified examples on the eventual robustness. They discover that misclassified examples significantly influence the final robustness and restructure the adversarial risk to include a distinct differentiation of misclassified examples through regularization. From the perspective of reweighting, CAT (Cai et al., 2018) reweights adversarial data with different PGD iterations $K$. DAT (Wang et al., 2019) reweights the adversarial data with different convergence qualities. More recently, Ding et al. (2020) proposes to reweight adversarial data with instance-dependent perturbation bounds $\epsilon$ and Zhang et al. (2021) proposes a geometry-aware instance-reweighted AT framework (GAIRAT) that assigns different weights to adversarial loss based on the distance of data points from the class boundary.

Our proposed method is fundamentally different from the existing methods. Existing reweighted AT methods primarily focus on instance-based reweighting, wherein each data instance is treated distinctly. Our proposed method, on the other hand, pioneers a pixel-based reweighting strategy, which allows for distinct treatment of pixel regions within each instance. Moreover, the design of PART is orthogonal to the state-of-the-art optimized AT methods such as TRADES (Zhang et al., 2019) and MART (Wang et al., 2020). This compatibility ensures that PART can be seamlessly integrated into these established frameworks, thereby extending its utility.

**Class activation mapping.** Vanilla CAM (Zhou et al., 2016) is designed for producing visual explanations of decisions made by CNN-based models by computing a coarse localization map highlighting important regions in an image for predicting a concept. Besides vanilla CAM, many improved CAM methods have been proposed. For example, GradCAM (Selvaraju et al., 2017) improves upon CAM by using the gradient information flowing into the last convolutional layer of the CNN to assign importance values to each neuron, enabling the production of class-discriminative visualizations without the need for architectural changes or re-training. XGradCAM (Fu et al., 2020) introduces two axioms to improve the sensitivity and conservation of GradCAM. Specifically, it uses

a modified gradient to better capture the importance of each feature map and a normalization term to preserve the spatial information of the feature maps. LayerCAM (Jiang et al., 2021) generates class activation maps not only from the final convolutional layer but also from shallow layers. This allows for both coarse spatial locations and fine-grained object details to be captured.

We want to make sure the chosen CAM method can truly reflect the importance of the pixel regions to avoid additional bias from the methods themselves. Therefore, we conduct experiments to compare the performance of PART with different CAM methods. We find that these state-of-the-art CAM methods have approximately identical performance (see Appendix K).

**Adversarial defenses with class activation mapping.** Zhou et al. (2021) proposes to use class activation features to remove adversarial noise. Specifically, it crafts AEs by maximally disrupting the class activation features of natural examples and then trains a denoising model to minimize the discrepancies between the class activation features of natural and AEs. Wu et al. (2023) proposes an Attention-based Adversarial Defense (AAD) framework that uses GradCAM to rectify and preserve the visual attention area, which aims to improve the robustness against adversarial attacks by aligning the visual attention area between adversarial and original images.

**Adversarial attacks with class activation mapping.** Dong et al. (2020) proposes an attack method that leverages superpixel segmentation and class activation mapping to focus on regions of an image that are most influential in classification decisions. It highlights the importance of considering perceptual features and classification-relevant regions in crafting effective AEs.

Our method differs from the above methods technically, which allocates varying perturbation budgets to different pixel regions. We want to emphasize that PART is a general idea rather than a specific method and CAM is one of the tools to realize our idea. The main goal of our work is to provide insights on how to design an effective AT method by counting the fundamental discrepancies of pixel regions across images.

# D    NOTATIONS IN SECTION 3.1

| | |
|---|---|
| $\ell$ | A loss function |
| $f$ | A model |
| $\mathbf{x}$ | A natural image |
| $y$ | The true label of $\mathbf{x}$ |
| $d$ | The data dimension |
| $\boldsymbol{\Delta}$ | The adversarial perturbation added to $\mathbf{x}$ |
| $\boldsymbol{\Delta}^*$ | The optimal solution of $\boldsymbol{\Delta}$ |
| $\|\|\cdot\|\|_\infty$ | The $\ell_\infty$-norm |
| $\epsilon$ | The maximum allowed perturbation budget for important pixels |
| $\epsilon^{\text{low}}$ | The maximum allowed perturbation budget for unimportant pixels |
| $\mathcal{I}^{\text{high}}$ | Indexes of important pixels |
| $\mathcal{I}^{\text{low}}$ | Indexes of unimportant pixels |
| $\boldsymbol{v}$ | A function to transform a set to a vector |
| $\{\delta_i\}_{i \in \mathcal{I}^{\text{high}}}$ | A set consisting of important pixels in $\boldsymbol{\Delta}$, i.e., $\boldsymbol{\Delta}^{\text{high}}$ |
| $\{\delta_i\}_{i \in \mathcal{I}^{\text{low}}}$ | A set consisting of unimportant pixels in $\boldsymbol{\Delta}$, i.e., $\boldsymbol{\Delta}^{\text{low}}$ |
| $|\mathcal{I}^{\text{high}}|$ | The dimension of important pixel regions, i.e., $d^{\text{high}}$ |
| $|\mathcal{I}^{\text{low}}|$ | The dimension of unimportant pixel regions, i.e., $d^{\text{low}}$ |

# E  ALGORITHMS

---

**Algorithm 1** Mask Generation

---

**Input:** data dimension $d$, normalized class activation map $\tilde{L} = [\omega_1, ..., \omega_d]$, maximum allowed perturbation budgets $\epsilon$, $\epsilon^{\text{low}}$
**Output:** mask $m$
 1: Initialize mask $m = \{m_1, ..., m_d\} = \mathbf{1}_d$
 2: **for** $i = 1, ..., d$ **do**
 3:     **if** $\omega_i > 1$ **then**
 4:         $m_i = \epsilon^{\text{low}}/\epsilon$
 5:     **end if**
 6: **end for**

---

**Algorithm 2** Pixel-reweighted AE Generation (PRAG)

---

**Input:** data $\mathbf{x} \in \mathcal{X}$, label $y \in \mathcal{Y}$, model $f$, loss function $\ell$, step size $\alpha$, number of iterations $K$ for inner optimization, maximum allowed perturbation budget $\epsilon$
**Output:** adversarial example $\tilde{\mathbf{x}}$
 1: Obtain mask $m$ by Algorithm 1
 2: $\tilde{\mathbf{x}} \leftarrow \mathbf{x}$
 3: **for** $k = 1, ..., K$ **do**
 4:     $\tilde{\mathbf{x}} \leftarrow \tilde{\mathbf{x}} + m \odot \text{clip}(\tilde{\mathbf{x}} + \alpha \text{sign}(\nabla_{\tilde{\mathbf{x}}} \ell(f(\tilde{\mathbf{x}}), y)) - \mathbf{x}, -\epsilon, \epsilon)$
 5: **end for**

---

**Algorithm 3** Pixel-reweighted Adversarial Training (PART)

---

**Input:** network $f$ with parameters $\boldsymbol{\theta}$, training dataset $S = \{(\mathbf{x}_i, y_i)\}_{i=1}^n$, learning rate $\eta$, number of epochs $T$, batch size $n$, numebr of batches $N$
**Output:** Robust network $f$
 1: **for** epoch $= 1, ..., T$ **do**
 2:     **for** mini-batch $= 1, ..., N$ **do**
 3:         Read mini-batch $B = \{\mathbf{x}_1, ..., \mathbf{x}_n\}$ from $S$
 4:         **for** $i = 1, ..., n$ (in parallel) **do**
 5:             Obtain adversarial data $\tilde{\mathbf{x}}_i$ of $\mathbf{x}_i$ by Algorithm 2
 6:         **end for**
 7:         $\boldsymbol{\theta} \leftarrow \boldsymbol{\theta} - \eta \sum_{i=1}^m \nabla_{\boldsymbol{\theta}} \ell(f(\tilde{\mathbf{x}}_i), y_i)$
 8:     **end for**
 9: **end for**

---

# F  PROOF OF THEOREM 1

**Problem settings.** Consider a 2D data point $\mathbf{x} = [x_1, x_2]^T$ with label $y$ and an adversarial perturbation $\boldsymbol{\Delta} = [\delta_1, \delta_2]^T$ that is added to $\mathbf{x}$, with $\delta_1 \in [-\epsilon_1, \epsilon_1]$ and $\delta_2 \in [-\epsilon_2, \epsilon_2]$. We consider a linear model $f(\mathbf{x}) = \omega_1 x_1 + \omega_2 x_2 + b$ for this problem, where $\omega_1$ and $\omega_2$ are the weights for pixels $x_1$ and $x_2$ respectively. We use the square loss here as it is differentiable, which can be expressed as $\ell(f(\mathbf{x}), y) = (y - f(\mathbf{x}))^2$. The objective of our problem is to find $\boldsymbol{\Delta}$ that can maximize $\ell(f(\mathbf{x} + \boldsymbol{\Delta}), y)$, which is equivalent to minimizing its negative counterpart. Thus, the constraint optimization problem can be formulated as follows:

$$
\begin{aligned}
\text{minimize} \quad & -(y - f(\mathbf{x} + \boldsymbol{\Delta}))^2, \\
\text{subject to} \quad & \delta_1 \leq \epsilon_1, -\delta_1 \leq \epsilon_1, \delta_2 \leq \epsilon_2, -\delta_2 \leq \epsilon_2.
\end{aligned}
\tag{17}
$$

By using Lagrange multiplier method, we can construct the following Lagrange function $\mathcal{L}$:

$$
\mathcal{L} = -(y - f(\mathbf{x} + \boldsymbol{\Delta}))^2 + \lambda_1(\delta_1 - \epsilon_1) + \lambda_2(-\delta_1 - \epsilon_1) + \lambda_3(\delta_2 - \epsilon_2) + \lambda_4(-\delta_2 - \epsilon_2). \tag{18}
$$

Expanding $\mathcal{L}$, we get:

$$
\begin{aligned}
\mathcal{L} = & -y^2 + 2y\omega_1 x_1 + 2y\omega_1\delta_1 + 2y\omega_2 x_2 + 2y\omega_2\delta_2 + 2yb - \omega_1^2 x_1^2 - 2\omega_1^2 x_1\delta_1 \\
& - 2\omega_1\omega_2 x_1 x_2 - 2\omega_1\omega_2 x_1\delta_2 - 2\omega_1 x_1 b - \omega_1^2\delta_1^2 - 2\omega_1\omega_2 x_2\delta_1 - 2\omega_1\omega_2\delta_1\delta_2 \\
& - 2\omega_1\delta_1 b - \omega_2^2 x_2^2 - 2\omega_2^2 x_2\delta_2 - 2\omega_2 x_2 b - \omega_2^2\delta_2^2 - 2\omega_2\delta_2 b - b^2 \\
& + \lambda_1\delta_1 - \lambda_1\epsilon_1 - \lambda_2\delta_1 - \lambda_2\epsilon_1 + \lambda_3\delta_2 - \lambda_3\epsilon_2 - \lambda_4\delta_2 - \lambda_4\epsilon_2.
\end{aligned}
\tag{19}
$$

Taking the derivatives with respect to $\delta_1$ and $\delta_2$ and setting them to zero, we have:

$$
\frac{\partial\mathcal{L}}{\partial\delta_1} = 2y\omega_1 - 2\omega_1^2 x_1 - 2\omega_1^2\delta_1 - 2\omega_1\omega_2 x_2 - 2\omega_1\omega_2\delta_2 - 2\omega_1 b + \lambda_1 - \lambda_2 = 0.
\tag{20}
$$

$$
\frac{\partial\mathcal{L}}{\partial\delta_2} = 2y\omega_2 - 2\omega_2^2 x_2 - 2\omega_2^2\delta_2 - 2\omega_1\omega_2 x_1 - 2\omega_1\omega_2\delta_1 - 2\omega_2 b + \lambda_3 - \lambda_4 = 0.
\tag{21}
$$

Solving Eq. (20) and Eq. (21), we can get the expressions for $\lambda_1^*$, $\lambda_2^*$, $\lambda_3^*$ and $\lambda_4^*$:

$$
\lambda_1^* = 2\omega_1^2 x_1 + 2\omega_1^2\delta_1^* + 2\omega_1\omega_2 x_2 + 2\omega_1\omega_2\delta_2^* + 2\omega_1 b - 2y\omega_1 + \lambda_2^*.
\tag{22}
$$

$$
\lambda_2^* = 2y\omega_1 - 2\omega_1^2 x_1 - 2\omega_1^2\delta_1^* - 2\omega_1\omega_2 x_2 - 2\omega_1\omega_2\delta_2^* - 2\omega_1 b + \lambda_1^*.
\tag{23}
$$

$$
\lambda_3^* = 2\omega_2^2 x_2 + 2\omega_2^2\delta_2^* + 2\omega_1\omega_2 x_1 + 2\omega_1\omega_2\delta_1^* + 2\omega_2 b - 2y\omega_2 + \lambda_4^*.
\tag{24}
$$

$$
\lambda_4^* = 2y\omega_2 - 2\omega_2^2 x_2 - 2\omega_2^2\delta_2^* - 2\omega_1\omega_2 x_1 - 2\omega_1\omega_2\delta_1^* - 2\omega_2 b + \lambda_3^*.
\tag{25}
$$

This is based on the *Karush–Kuhn–Tucker* (KKT) conditions (Avriel, 2003):

$$
\delta_1^* \le \epsilon_1, -\delta_1^* \le \epsilon_1, \delta_2^* \le \epsilon_2, -\delta_2^* \le \epsilon_2.
\tag{26}
$$

$$
\lambda_1^* \ge 0, \lambda_2^* \ge 0, \lambda_3^* \ge 0, \lambda_4^* \ge 0.
\tag{27}
$$

$$
\lambda_1^*(\delta_1^* - \epsilon_1) = 0, \lambda_2^*(-\delta_1^* - \epsilon_1) = 0, \lambda_3^*(\delta_2^* - \epsilon_2) = 0, \lambda_4^*(-\delta_2^* - \epsilon_2) = 0.
\tag{28}
$$

Consider Eq. (28), we can further see two conditions:

1. $\lambda_1^*$ and $\lambda_2^*$ cannot be greater than 0 simultaneously. Otherwise $\delta_1^*$ equals to $\epsilon_1$ and $-\epsilon_1$ simultaneously. This only holds when $\epsilon_1 = -\epsilon_1 = 0$ which means there is no perturbation added to $x_1$, and thus breaks away from adversarial settings.

2. Similarly, $\lambda_3^*$ and $\lambda_4^*$ cannot be greater than 0 simultaneously.

Considering all the conditions, we can summarize the generated AEs into three cases:

1. When $\lambda_1^* = \lambda_2^* = \lambda_3^* = \lambda_4^* = 0$. If we substitute the values of $\lambda^*$s into Eq. (19), we can see all the terms related to $\epsilon_1$ and $\epsilon_2$ are eliminated. This means if we take the derivatives of Eq. (19) with respect to $\delta_1$ and $\delta_2$, the optimal $\delta_1^*$ and $\delta_2^*$ will be some expressions without $\epsilon_1$ and $\epsilon_2$. This means the optimized solutions are inside $(-\epsilon_1, \epsilon_1)$. If $\delta_1^*$ and $\delta_2^*$ are far from the boundary, moderately change $\epsilon$ would hardly affect the results.

2. When one of $\lambda_1^*$, $\lambda_2^*$ is greater than 0, and one of $\lambda_3^*$, $\lambda_4^*$ is greater than 0. Take $(\lambda_1^* > 0, \lambda_2^* = 0, \lambda_3^* > 0, \lambda_4^* = 0)$ as an example, both $\delta_1^*$ and $\delta_2^*$ reach the boundary condition Eq. (28), i.e., $\delta_1^* = \epsilon_1$ and $\delta_2^* = \epsilon_2$. If we substitute $\delta_1^* = \epsilon_1$ and $\delta_2^* = \epsilon_2$ and $\lambda^*$s into Eq. (18), we have:

$$
\mathcal{L} = -(y - f(\mathbf{x}) - \omega_1\epsilon_1 - \omega_2\epsilon_2)^2.
\tag{29}
$$

We can see the significance of $\epsilon_1$ and $\epsilon_2$ is different if $\omega_1 \ne \omega_2$.

3. When only one of $\lambda^*$s is greater than 0, while others are 0. Take $(\lambda_1^* > 0, \lambda_2^* = \lambda_3^* = \lambda_4^* = 0)$ as an example, then $\delta_1^* = \epsilon_1$ according to Eq. (28). If we substitute $\delta_1^* = \epsilon_1$ into Eq. (21), we can get:

$$
\delta_2^* = \frac{y - f(\mathbf{x}) - \omega_1\epsilon_1}{\omega_2}, \text{ subject to } \delta_1^* = \epsilon_1.
\tag{30}
$$

**Main takeaway.** If two pixels have different influences on the model's predictions, it will affect the generation process of AEs, leading to different solutions of the optimal $\delta^*$. Thus, it probably influences the performance of AT.

For completeness, we list the remaining cases as follows:

1. $(\lambda_1^* = 0, \lambda_2^* > 0, \lambda_3^* = 0, \lambda_4^* > 0)$. In this case, $\delta_1^* = -\epsilon_1$ and $\delta_2^* = -\epsilon_2$.

2. $(\lambda_1^* = 0, \lambda_2^* > 0, \lambda_3^* > 0, \lambda_4^* = 0)$. In this case, $\delta_1^* = -\epsilon_1$ and $\delta_2^* = \epsilon_2$.

3. $(\lambda_1^* > 0, \lambda_2^* = 0, \lambda_3^* = 0, \lambda_4^* > 0)$. In this case, $\delta_1^* = \epsilon_1$ and $\delta_2^* = -\epsilon_2$.

4. $(\lambda_2^* > 0, \lambda_1^* = \lambda_3^* = \lambda_4^* = 0)$, then $\delta_1^* = -\epsilon_1$ according to Eq. (28). If we substitute $\delta_1^* = -\epsilon_1$ into Eq. (21), we can get:

$$\delta_2^* = \frac{y - f(\mathbf{x}) + \omega_1 \epsilon_1}{\omega_2}, \text{ subject to } \delta_1^* = -\epsilon_1.$$

5. $(\lambda_3^* > 0, \lambda_1^* = \lambda_2^* = \lambda_4^* = 0)$, then $\delta_2^* = \epsilon_2$ according to Eq. (28). If we substitute $\delta_2^* = \epsilon_2$ into Eq. (20), we can get:

$$\delta_1^* = \frac{y - f(\mathbf{x}) - \omega_2 \epsilon_2}{\omega_1}, \text{ subject to } \delta_1^* = \epsilon_2.$$

6. $(\lambda_4^* > 0, \lambda_1^* = \lambda_2^* = \lambda_3^* = 0)$, then $\delta_2^* = -\epsilon_2$ according to Eq. (28). If we substitute $\delta_2^* = -\epsilon$ into Eq. (20), we can get:

$$\delta_1^* = \frac{y - f(\mathbf{x}) + \omega_2 \epsilon_2}{\omega_1}, \text{ subject to } \delta_1^* = -\epsilon_2.$$

**Remark.** Note that, we do not cover how different levels of pixel importance would affect the performance of AT. This is because, during AT, the generated AEs are highly correlated, making the training process quite complicated to analyze in theory. According to recent developments regarding learning with dependent data (Dagan et al., 2019), we can only expect generalization when weak dependence exists in training data. However, after the first training epoch in AT, the model already depends on all training data, meaning that the generated AEs in the following epochs are probably highly dependent on each other. Thus, we leave this as our future work.

## G   DETAILED EXPERIMENT SETTINGS

**Dataset.** We evaluate the effectiveness of PART on three benchmark datasets, i.e., CIFAR-10 (Krizhevsky et al., 2009), SVHN (Netzer et al., 2011) and Tiny-ImageNet (Wu, 2017). CIFAR-10 comprises 50,000 training and 10,000 test images, distributed across 10 classes, with a resolution of $32 \times 32$. SVHN has 10 classes but consists of 73,257 training and 26,032 test images, maintaining the same $32 \times 32$ resolution. Tiny-ImageNet extends the complexity by offering 200 classes with a higher resolution of $64 \times 64$, containing 100,000 training, 10,000 validation, and 10,000 test images. For the target model, following the idea in Zhou et al. (2023), we use ResNet (He et al., 2016) for CIFAR-10 and SVHN, and WideResNet (Zagoruyko & Komodakis, 2016) for Tiny-ImageNet.

**Attack settings.** We mainly use three types of adversarial attacks to evaluate the performances of defenses. They are $\ell_\infty$-norm PGD (Madry et al., 2018), $\ell_\infty$-norm MMA (Gao et al., 2022) and $\ell_\infty$-norm AA (Croce & Hein, 2020a). Among them, AA is a combination of three non-target white-box attacks (Croce & Hein, 2020b) and one targeted black-box attack (Andriushchenko et al., 2020), which makes AA a gold standard for evaluating adversarial robustness up to this point. Recently proposed MMA (Gao et al., 2022) can achieve comparable performance compared to AA but is much more time efficient. The iteration number for PGD is set to 20 (Zhou et al., 2023), and the target selection number for MMA is set to 3 (Gao et al., 2022), respectively. For all attacks, we set $\epsilon$ to 8/255.

**Defense settings.** We use three representative AT methods as the baselines: AT (Madry et al., 2018) and two optimized AT methods TRADES (Zhang et al., 2019) and MART (Wang et al., 2020). We set $\lambda = 6$ for both TRADES and MART. For all baseline methods, we use the $\ell_\infty$-norm non-targeted PGD-10 with random start to craft AEs in the training stage. We set $\epsilon = 8/255$ for all datasets, and $\epsilon^{\text{low}} = 7/255$ for our method. All the defense models are trained using SGD with a momentum of 0.9. Following Zhou et al. (2023) and Gao et al. (2022), we set the initial learning rate to 0.01 with batch size 128 for CIFAR-10 and SVHN. To save time, we set the initial learning rate to 0.02 with batch size 512 for Tiny-ImageNet. The step size $\alpha$ is 2/255 for CIFAR-10 and Tiny-ImageNet, and is 1/255 for SVHN. The weight decay is 0.0002 for CIFAR-10, 0.0035 for SVHN and 0.0005 for Tiny-ImageNet. We run all the methods for 80 epochs and divide the learning rate by 10 at epoch 60 to avoid robust overfitting (Rice et al., 2020). In PART, the initial 20 epochs is the burn-in period.

## H    POSSIBILITY OF OBFUSCATED GRADIENTS

We consider the five behaviours listed in Athalye et al. (2018) to identify the obfuscated gradients:

1. We find that *one-step attacks do not perform better than iterative attacks*. The accuracy of our method against PGD-1 is 76.31% (vs 43.65% against PGD-20).

2. We find that *black-box attacks have lower attack success rates than white-box attacks*. We use ResNet-18 with AT as the surrogate model to generate AEs. The accuracy of our method against PGD-20 is 59.17% (vs 43.65% in the white-box setting).

3. We find that *unbounded attacks reach 100% success*. The accuracy of our method against PGD-20 with $\epsilon = 255/255$ is 0%.

4. We find that *random sampling does not find AEs*. For samples that are not successfully attacked by PGD, we randomly sample 100 points within the $\epsilon$-ball and do not find adversarial data.

5. We find that *increasing distortion bound increases success*. The accuracy of our method against PGD-20 with increasing $\epsilon$ (8/255, 16/255, 32/255 and 64/255) is 43.65%, 10.70%, 0.49% and 0%.

These results show that our method does not cause obfuscated gradients.

## I    ADDITIONAL EXPERIMENTS ON THE IMPACT OF ATTACK ITERATIONS

Table 3: Robustness (%) of defense methods against PGD with different iterations on CIFAR-10. We report the averaged results and standard deviations of three runs. We show the most successful defense in **bold**.

| Dataset | Method | ResNet-18 | | | | |
| --- | --- | --- | --- | --- | --- | --- |
| | | PGD-10 | PGD-40 | PGD-60 | PGD-80 | PGD-100 |
| CIFAR-10 | AT | $44.83 \pm 0.13$ | $43.00 \pm 0.10$ | $42.83 \pm 0.07$ | $42.81 \pm 0.03$ | $42.81 \pm 0.03$ |
| | PART | $\mathbf{45.20 \pm 0.17}$ | $\mathbf{43.20 \pm 0.14}$ | $\mathbf{43.09 \pm 0.09}$ | $\mathbf{43.08 \pm 0.10}$ | $\mathbf{42.93 \pm 0.07}$ |
| | TRADES | $48.81 \pm 0.21$ | $48.19 \pm 0.13$ | $48.16 \pm 0.15$ | $48.14 \pm 0.08$ | $48.08 \pm 0.04$ |
| | PART-T | $\mathbf{49.41 \pm 0.11}$ | $\mathbf{48.65 \pm 0.10}$ | $\mathbf{48.64 \pm 0.13}$ | $\mathbf{48.64 \pm 0.04}$ | $\mathbf{48.62 \pm 0.03}$ |
| | MART | $49.98 \pm 0.08$ | $49.66 \pm 0.16$ | $49.66 \pm 0.06$ | $49.54 \pm 0.03$ | $49.47 \pm 0.05$ |
| | PART-M | $\mathbf{50.50 \pm 0.19}$ | $\mathbf{50.19 \pm 0.15}$ | $\mathbf{50.09 \pm 0.04}$ | $\mathbf{50.06 \pm 0.05}$ | $\mathbf{50.05 \pm 0.02}$ |

Table 4: Robustness (%) and Accuracy (%) of PART against PGD with different iterations during training on CIFAR-10. The target model is ResNet-18. We report the averaged results and standard deviations of three runs.

| Dataset | Method | ResNet-18 | | | |
| --- | --- | --- | --- | --- | --- |
| | | Natural | PGD-20 | MMA | AA |
| CIFAR-10 | PART (PGD-10) | $83.42 \pm 0.26$ | $43.65 \pm 0.16$ | $41.98 \pm 0.03$ | $41.74 \pm 0.04$ |
| | PART (PGD-20) | $83.44 \pm 0.19$ | $43.64 \pm 0.13$ | $42.02 \pm 0.13$ | $41.82 \pm 0.08$ |
| | PART (PGD-40) | $83.36 \pm 0.21$ | $43.82 \pm 0.08$ | $42.09 \pm 0.07$ | $41.86 \pm 0.11$ |
| | PART (PGD-60) | $83.30 \pm 0.15$ | $44.02 \pm 0.12$ | $42.18 \pm 0.05$ | $41.91 \pm 0.09$ |

We conduct extra experiments to analyze the impact of attack iterations on the performance of CAM methods. Specifically, we test the robustness of defense methods against PGD with different iterations on CIFAR-10 (see Table 3). With the increase of attack iterations, the robustness of defense methods will decrease. This is because the possibility of finding worst-case examples will increase with more attack iterations. The effectiveness of CAM technology itself, however, is rarely influenced by attack iterations, as our method can consistently outperform baseline methods.

Furthermore, we take a close look at how the number of attack iterations during training would affect the final performance of CAM methods (see Table 4). Similarly, if we increase the attack iterations

during training, the model will become more robust as the model learns more worst-case examples during training. At the same time, the natural accuracy has a marginal decrease. Overall, we can obtain same conclusions that the performance of our method is stable and CAM methods are rarely affected by the attack iterations.

## J   ADDITIONAL EXPERIMENT ON ADAPTIVE MMA ATTACK

Table 5: Robustness (%) of defense methods against adaptive MMA on CIFAR-10. We report the averaged results and standard deviations of three runs. We show the most successful defense in **bold**.

| Dataset | Method | ResNet-18 | | | | |
| | | MMA-20 | MMA-40 | MMA-60 | MMA-80 | MMA-100 |
| --- | --- | --- | --- | --- | --- | --- |
| CIFAR-10 | AT | $35.36 \pm 0.10$ | $35.02 \pm 0.05$ | $34.93 \pm 0.09$ | $34.86 \pm 0.06$ | $34.85 \pm 0.07$ |
| | PART | $\mathbf{35.67 \pm 0.07}$ | $\mathbf{35.35 \pm 0.11}$ | $\mathbf{35.29 \pm 0.13}$ | $\mathbf{35.29 \pm 0.09}$ | $\mathbf{35.17 \pm 0.05}$ |
| | TRADES | $40.14 \pm 0.08$ | $39.89 \pm 0.12$ | $39.93 \pm 0.05$ | $39.87 \pm 0.08$ | $39.82 \pm 0.03$ |
| | PART-T | $\mathbf{40.78 \pm 0.13}$ | $\mathbf{40.57 \pm 0.11}$ | $\mathbf{40.51 \pm 0.08}$ | $\mathbf{40.49 \pm 0.05}$ | $\mathbf{40.48 \pm 0.02}$ |
| | MART | $39.14 \pm 0.06$ | $38.79 \pm 0.13$ | $38.80 \pm 0.10$ | $38.79 \pm 0.05$ | $38.74 \pm 0.08$ |
| | PART-M | $\mathbf{40.56 \pm 0.11}$ | $\mathbf{40.26 \pm 0.07}$ | $\mathbf{40.23 \pm 0.12}$ | $\mathbf{40.21 \pm 0.08}$ | $\mathbf{40.20 \pm 0.07}$ |

For adaptive attacks, we conduct an additional experiment to test the robustness of defense methods against adaptive MMA (see Table 5 above). The choice of MMA over AA for adaptive attacks is due to AA's time-consuming nature as an ensemble of multiple attacks. Incorporating the CAM method into AA would further slow the process. MMA, in contrast, offers greater time efficiency and comparable performance to AA.

## K   ADDITIONAL EXPERIMENT ON DIFFERENT CAM METHODS

Table 6: Comparison of PART's performance with different CAM methods on CIFAR-10. We report the averaged results and standard deviations of three runs.

| Method | CAM | ResNet-18 (CIFAR-10) | | | |
| | | Natural | PGD-20 | MMA | AA |
| --- | --- | --- | --- | --- | --- |
| PART | GradCAM | $83.42 \pm 0.26$ | $43.65 \pm 0.06$ | $41.98 \pm 0.03$ | $41.74 \pm 0.04$ |
| | XGradCAM | $83.34 \pm 0.18$ | $43.53 \pm 0.08$ | $41.97 \pm 0.05$ | $41.74 \pm 0.02$ |
| | LayerCAM | $83.38 \pm 0.21$ | $43.67 \pm 0.11$ | $42.07 \pm 0.09$ | $42.03 \pm 0.16$ |

## L   ADDITIONAL EXPERIMENT ON DIFFERENT AE GENERATION METHODS

Table 7: Comparison of PART's performance with different CAM methods on CIFAR-10. We report the averaged results and standard deviations of three runs.

| AE Generation | Method | ResNet-18 (CIFAR-10) | | | |
| | | Natural | PGD-20 | MMA | AA |
| --- | --- | --- | --- | --- | --- |
| PGD-10 | AT | $82.58 \pm 0.05$ | $\mathbf{43.69 \pm 0.28}$ | $41.80 \pm 0.10$ | $41.63 \pm 0.22$ |
| | PART | $\mathbf{83.42 \pm 0.26}$ | $43.65 \pm 0.06$ | $\mathbf{41.98 \pm 0.03}$ | $\mathbf{41.74 \pm 0.04}$ |
| MMA | AT | $81.76 \pm 0.11$ | $44.76 \pm 0.14$ | $42.31 \pm 0.13$ | $42.04 \pm 0.15$ |
| | PART | $\mathbf{83.55 \pm 0.28}$ | $\mathbf{44.99 \pm 0.14}$ | $\mathbf{42.50 \pm 0.22}$ | $\mathbf{42.09 \pm 0.24}$ |

# M    EXTRA COST INTRODUCED BY CAM METHODS

To avoid introducing unaffordable computational time by CAM methods, we update the mask $m$ for every 10 epochs. We show that the performance of our method remains competitive (see Table 8) given the mask is updated for every 10 epochs. Regarding memory consumption, the majority of the memory is allocated for storing checkpoints, with only a small portion attributed to CAM technology. We compare the computational time (hours : minutes : seconds) and the memory consumption (MB) of our method to different AT methods. See Table 9 and 10 for more details.

Table 8: Robustness (%) of defense methods on CIFAR-10. The target model is ResNet-18. We report the averaged results and standard deviations of three runs. We show the most successful defense in **bold**.

| Method | ResNet-18 (CIFAR-10) | | | |
| | Natural | PGD-20 | MMA | AA |
|---|---|---|---|---|
| AT | $82.58 \pm 0.14$ | $\mathbf{43.69 \pm 0.28}$ | $41.80 \pm 0.10$ | $41.63 \pm 0.22$ |
| PART (update $m$ every epoch) | $83.42 \pm 0.26$ | $43.65 \pm 0.16$ | $\mathbf{41.98 \pm 0.03}$ | $\mathbf{41.74 \pm 0.04}$ |
| PART (update $m$ every 10 epochs) | $\mathbf{83.77 \pm 0.15}$ | $43.36 \pm 0.21$ | $41.83 \pm 0.07$ | $41.41 \pm 0.14$ |
| TRADES | $78.16 \pm 0.15$ | $48.28 \pm 0.05$ | $45.00 \pm 0.08$ | $45.05 \pm 0.12$ |
| PART-T (update $m$ every epoch) | $79.36 \pm 0.31$ | $\mathbf{48.90 \pm 0.14}$ | $\mathbf{45.90 \pm 0.07}$ | $\mathbf{45.97 \pm 0.06}$ |
| PART-T (update $m$ every 10 epochs) | $\mathbf{80.13 \pm 0.16}$ | $48.72 \pm 0.11$ | $45.59 \pm 0.09$ | $45.60 \pm 0.04$ |
| MART | $76.82 \pm 0.28$ | $49.86 \pm 0.32$ | $45.42 \pm 0.04$ | $45.10 \pm 0.06$ |
| PART-M (update $m$ every epoch) | $78.67 \pm 0.10$ | $\mathbf{50.26 \pm 0.17}$ | $\mathbf{45.53 \pm 0.05}$ | $\mathbf{45.19 \pm 0.04}$ |
| PART-M (update $m$ every 10 epochs) | $\mathbf{80.00 \pm 0.15}$ | $49.71 \pm 0.12$ | $45.14 \pm 0.10$ | $44.61 \pm 0.24$ |

Table 9: Computational time (hours : minutes : seconds) of defense methods on CIFAR-10.

| | ResNet-18 (CIFAR-10) | | |
| GPU | Method | Training Speed | Difference |
|---|---|---|---|
| 1*NVIDIA A100 | SAT | 02:14:37 | 00:29:08 |
| | PART | 02:43:45 | |
| | TRADES | 02:44:19 | 00:30:47 |
| | PART-T | 03:15:06 | |
| | MART | 02:09:23 | 00:30:14 |
| | PART-M | 02:39:37 | |

Table 10: Memory consumption (MB) of defense methods on CIFAR-10.

| | ResNet-18 (CIFAR-10) | |
| Method | Memory Consumption | Difference |
|---|---|---|
| SAT | 5530MB | 347MB |
| PART | 5877MB | |
| TRADES | 5369MB | 319MB |
| PART-T | 5688MB | |
| MART | 5553MB | 341MB |
| PART-M | 5894MB | |

# N    ADDITIONAL EVIDENCE

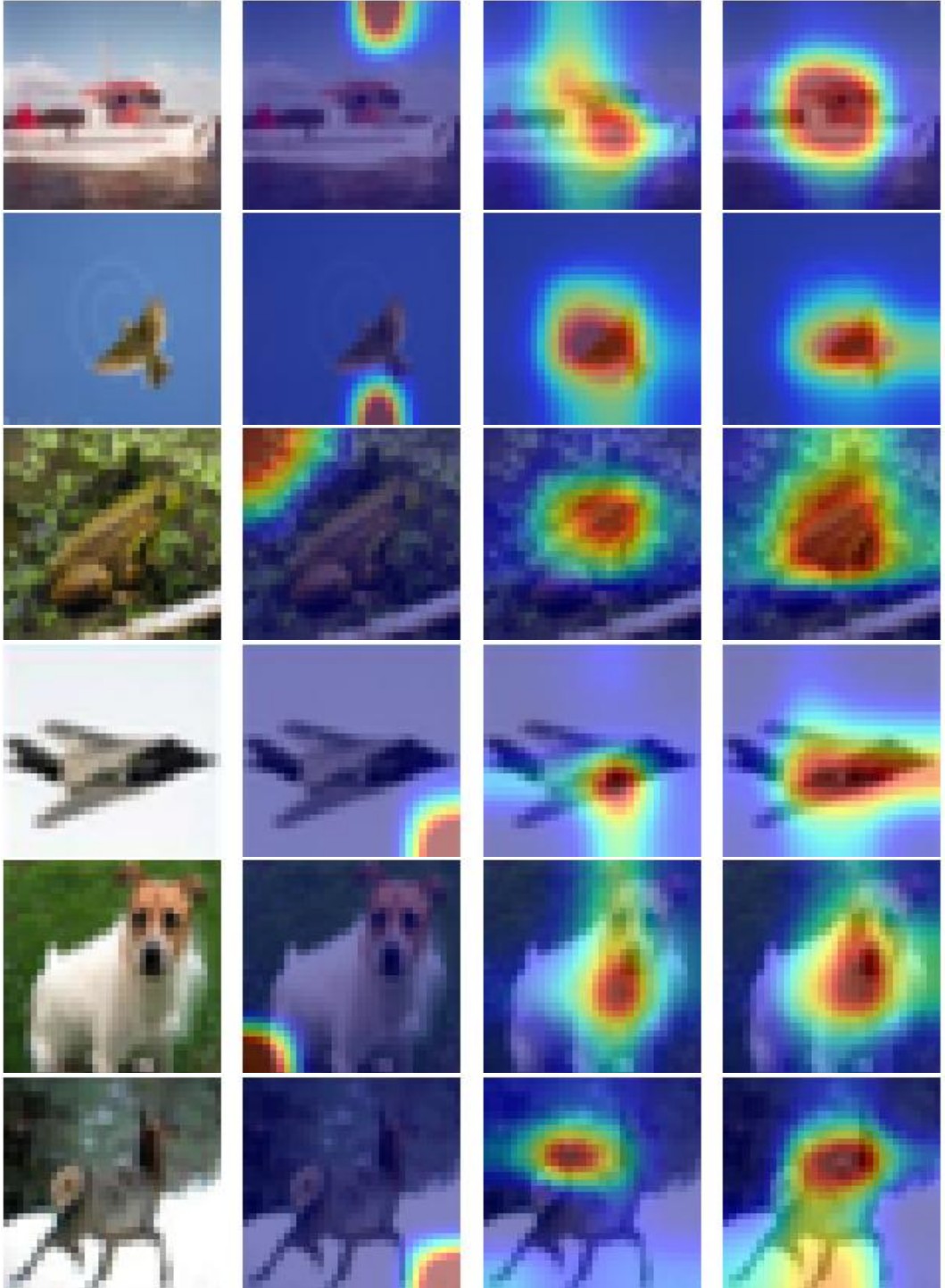

Figure 6: The examples of the high-contribution pixel regions learnt by ST, AT and PART. The first column contains original images. The second-to-last columns show the *important pixel regions* learnt by ST, AT and PART respectively on CIFAR-10.

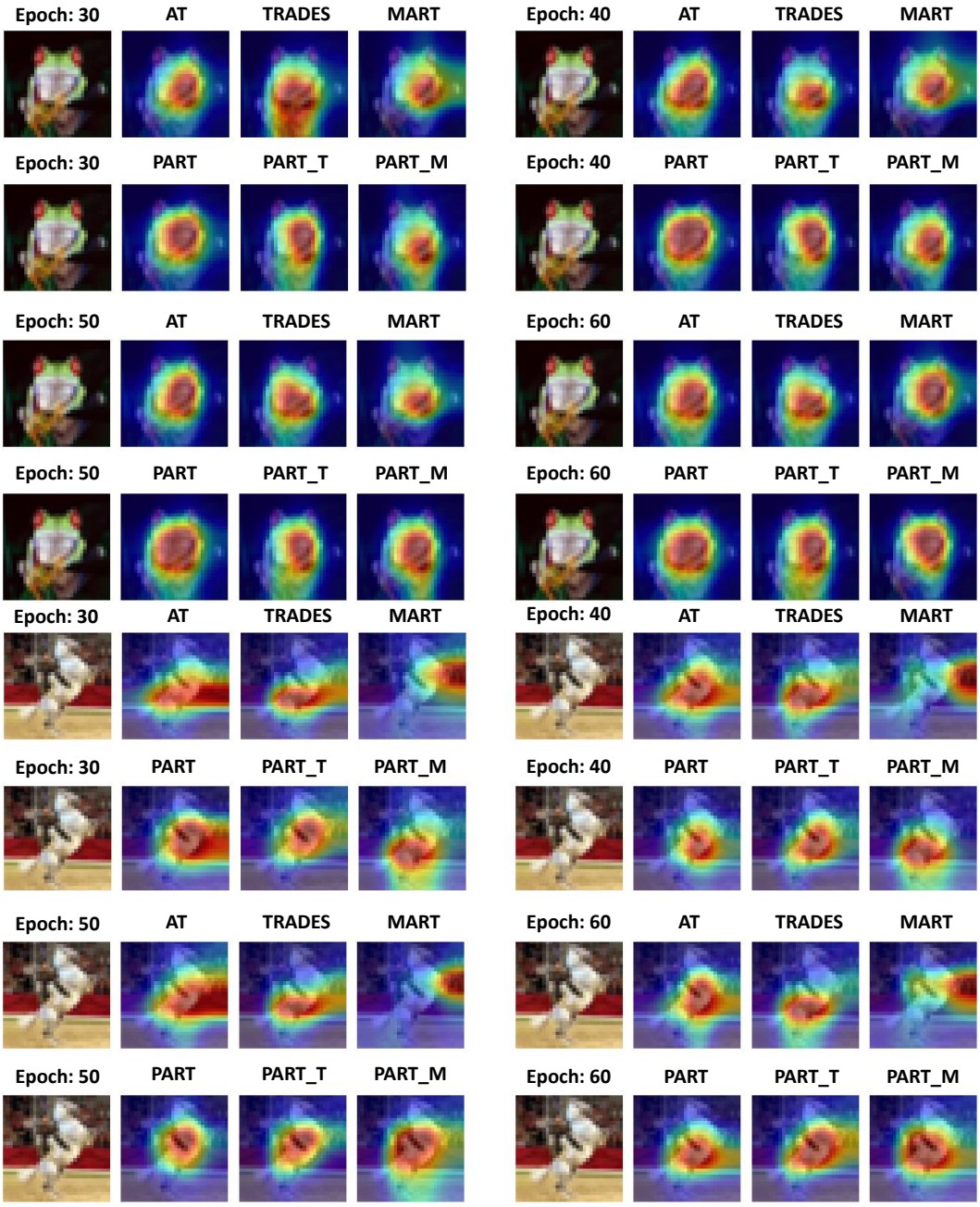

Figure 7: The additional examples of how the high-contribution pixel regions change with epoch number $\in \{30, 40, 50, 60\}$ on CIFAR-10.

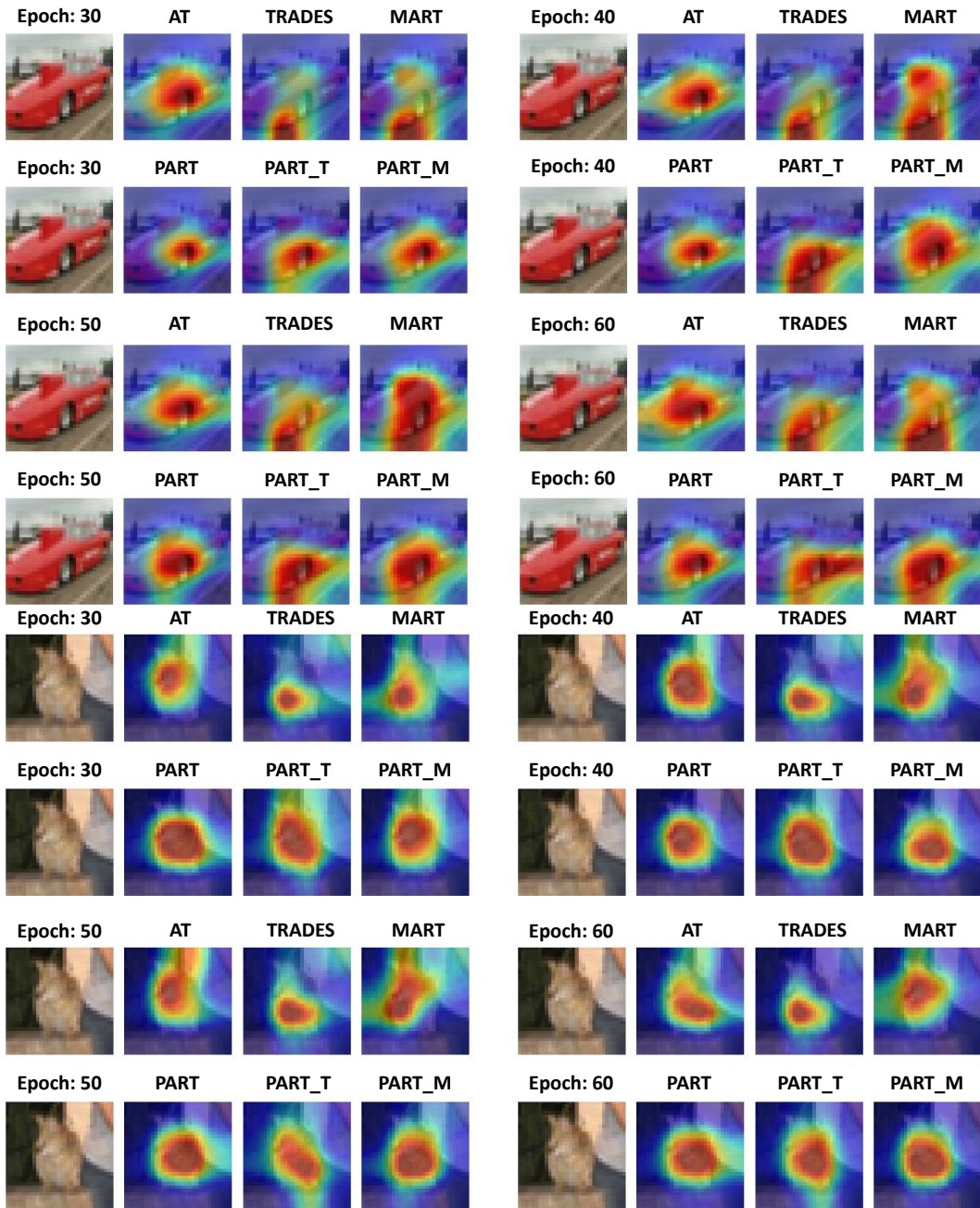

Figure 8: The additional examples of how the high-contribution pixel regions change with epoch number $\in \{30, 40, 50, 60\}$ on CIFAR-10.

