# OpenReview forum: "Pixel Reweighted Adversarial Training"
_ICLR.cc/2024/Conference — Submitted to ICLR 2024_

### Official Review · Reviewer_h86N · 2023-10-22

**Soundness:** 2 fair
**Presentation:** 3 good
**Contribution:** 2 fair
**Rating:** 5
**Confidence:** 4

**Summary:**

This paper introduces Pixel-reweighted AdveRsarial Training (PART), a novel framework within Adversarial Training (AT). It focuses on optimizing the perturbation budget ε by assigning higher allocations to pixels crucial for the model's performance and lower allocations to less influential ones. PART shows improved model robustness and accuracy compared to existing defenses. This performance enhancement is observed across datasets such as CIFAR-10, SVHN, and Tiny-ImageNet, even in the face of diverse attack strategies, including adaptive ones.

**Strengths:**

The paper introduces an original approach in Adversarial Training (AT) by addressing perturbation budgets and proposing Pixel-reweighted AdveRsarial Training (PART), offering a fresh perspective on adversarial defense. It maintains a solid research quality with a well-defined methodology, rigorous experiments, and comprehensive result documentation. Besides, this paper effectively communicates its problem formulation, methodology, and results, ensuring clear presentation.

**Weaknesses:**

1. The introduced CAM technology seems to have a weak improvement in robustness, and the authors did not analyze the impact on the training speed of the original AT framework, so I think it is not clear whether the performance trade-off brought by CAM is worth the cost of training speed and memory;
2. The authors mentioned that in the process of AT, it actually needs to be combined with standard AT for warm-up. I think they should specify the number of training sessions required for AT and PRAG during training;
3. The CAM technique seems to be a visualization technique for problems based on classification, which may limit the applicability of the PART framework proposed by the authors on other applications besides classification.
4. The authors’ description of the results in the figures and tables is not clear enough. For example, some of the tables seem to have standard deviations, while some do not. The authors didn't mention how many runs the standard deviation was calculated; The typography of the font, shadows, and content in the table are not compact enough.
5. Authors should probably consider more AT methods using CAM techniques mentioned in related works for comparison, instead of using many experimental results to compare different CAM techniques, since these techniques including naive CAM techniques seem to be existing
6. The authors do not seem to consider the impact of the number of attack iterations on the robustness in the results of white-box attack defense, such as the performance of PGD-10/50 and other attacks. In fact, the number of attacks may also have an important impact on the role of CAM technology.

**Questions:**

1. Regarding the use of CAM technology, can the authors provide a more detailed analysis of its impact on training speed and memory consumption when integrated with the original AT framework? Are the performance gains achieved through CAM technology worth the potential trade-offs?
2. The paper mentions combining standard AT for warm-up during training. Could the authors specify the number of training sessions required for both AT and PRAG in this process? How does this affect the training performance?
3. CAM technology appears to be a visualization technique primarily applicable to classification problems. How can the authors address concerns about the limited applicability of the PART framework in domains beyond classification? Are there plans to extend its use to other areas?
4. The clarity of results presentation is a concern. Could the authors provide more information about how standard deviations were calculated and specify the number of runs used in this calculation?
5. Instead of comparing various CAM techniques, could the authors consider comparing the PART framework with a broader range of AT methods that utilize CAM technology, thereby offering a more comprehensive comparison of its effectiveness?
6. White-box attack defense results are discussed, but the impact of the number of attack iterations on the performance of CAM technology isn't addressed. Can the authors provide insights into how the number of attack iterations influences the effectiveness of CAM technology in adversarial defense?

---

> ### Author Response · Authors · 2023-11-19
> **Response to Reviewer h86N (part 1)**
>
> Thank you so much for your time and efforts spent on reviewing our paper! Your thorough main review and comments are very important to the improvement of our work! Please find our replies below.
>
> >Q1. Regarding the use of CAM technology, can the authors provide a more detailed analysis of its impact on training speed and memory consumption when integrated with the original AT framework?
>
> A1: Incorporating CAM technologies will naturally lead to an increase in training speed. How to effectively integrate these technologies into the training process presents an intriguing challenge. If we generate the mask m for each epoch, the computational cost will be considerably high. **To address this problem, we update the mask m every 10 epochs.** We show that the performance of our method remains competitive (see Table 1) given the mask is updated for every 10 epochs. Regarding memory consumption, the majority of the memory is allocated for storing checkpoints, with only a small portion attributed to CAM technology. In general, the additional costs (i.e., training speed and memory consumption) are affordable (see Table 2 and 3). **This information has been included in the updated version of our manuscript.** For more details, please refer to Appendix M in the updated version.
>
> Table 1: Robustness (\%) of defense methods on CIFAR-10. The target model is ResNet-18. We report the averaged results and standard deviations of three runs. We show the most successful defense in **bold**.
>
> | Method | Natural     | PGD-20      | MMA         | AA          |
> |--------|-------------|-------------|-------------|-------------|
> | AT     | 82.58 ± 0.14| **43.69 ± 0.28**| 41.80 ± 0.10| 41.63 ± 0.22|
> | PART (update m for every epoch) | 83.42 ± 0.26 | 43.65 +- 0.16 | **41.98 ± 0.03** | **41.74 ± 0.04** |
> | PART (update m for every 10 epochs) | **83.77 ± 0.15** | 43.36 +- 0.21 | 41.83 ± 0.07 | 41.41 ± 0.14 |
> | TRADES | 78.16 ± 0.15| 48.28 ± 0.05| 45.00 ± 0.08| 45.05 ± 0.12|
> | PART-T (update m for every epoch) | 79.36 ± 0.31 | **48.90 ± 0.14**|**45.90 ± 0.07**| **45.97 ± 0.06** |\
> | PART-T (update m for every 10 epochs) | **80.13 ± 0.16** | 48.72 ± 0.11|45.59 ± 0.09|45.60 ± 0.04|
> | MART   | 76.82 ± 0.28 | 49.86 ± 0.32  | 45.42 ± 0.04| 45.10 ± 0.06 |
> | PART-M (update m for every epoch) |78.67 ± 0.10| **50.26 ± 0.17**|**45.53 ± 0.05** | **45.19 ± 0.04**|
> | PART-M (update m for every 10 epochs) |**80.00 ± 0.15**| 49.71 ± 0.12|45.14 ± 0.10| 44.61 ± 0.24|
>
> Table 2: Training speed (hours:minutes:seconds) of defense methods on CIFAR-10. The target model is ResNet-18.
> |GPU              |Method  | Training Speed | Difference |
> |-----------------|--------|----------------|------------|
> | 1 \* NVIDIA A100| SAT    | 02:14:37       | \          |
> |                 | PART   | 02:43:45       | 00:29:08   |
> |                 | TRADES | 02:44:19       | \          |
> |                 | PART-T | 03:15:06       | 00:30:47   |
> |                 | MART   | 02:09:23       | \          |
> |                 | PART-M | 02:39:37       | 00:30:14   |
>
>
> Table 3: Memory consumption (MB) of defense methods on CIFAR-10. The target model is ResNet-18.
> | Method   | Memory Consumption | Difference |
> |----------|--------------------|------------|
> | SAT      | 5530MB             |   \        |
> | PART     | 5877MB             | 347MB      |
> | TRADES   | 5369MB             |    \       |
> | PART-T   | 5688MB             | 319MB      |
> | MART     | 5553MB             |     \      |
> | PART-M   | 5894MB             | 341MB      |
>
>
> > Q2. Are the performance gains achieved through CAM technology worth the potential trade-offs?
>
> A2: From Table 1 and 3, we believe the extra cost is marginal compared to the performance gains. Besides, it is reasonable that our method gains marginal improvement in robustness, as PART decreases the attack strength for unimportant pixel regions. **What we want to highlight is that PART can improve the robustness-accuracy trade-off by notably increasing the natural accuracy.** Despite a marginal reduction in robustness by 0.04\% on PGD-20, PART gains more on natural accuracy (e.g., 2.08\% on SVHN and 1.36\% on Tiny-ImageNet). In most cases, PART can improve natural accuracy and robustness simultaneously. Compared to TRADES and MART, our method can still boost natural accuracy (e.g., 1.20\% on CIFAR-10, 2.64\% on SVHN and 1.26\% on Tiny-ImageNet for PART-T, and 1.85\% on CIFAR-10, 1.66\% on SVHN and 1.07\% on Tiny-ImageNet) with at most a 0.10\% drop in robustness. It is widely acknowledged that AT will hurt natural accuracy compared to standard training. **Our work can bridge this gap without compromising robustness.** As demonstrated in [1], advancements in the field of AT have been incremental over the past several years. As a result, making every slight improvement in this area is noteworthy.
>
> [1]  Rahul Rade and Seyed-Mohsen Moosavi-Dezfooli. Reducing excessive margin to achieve a better accuracy vs. robustness trade-off. In *ICLR*, 2022.

---

> ### Author Response · Authors · 2023-11-19
> **Response to Reviewer h86N (part 2)**
>
> Thank you so much for your time and efforts spent on reviewing our paper! Your thorough main review and comments are very important to the improvement of our work! Please find our replies below.
>
> > Q3. The paper mentions combining standard AT for warm-up during training. Could the authors specify the number of training sessions required for both AT and PRAG in this process?
>
> A3: **The warm-up epoch number for PART is 20.** Specifically, we use AT to train the model for the first 20 epochs and then use PART for the remaining 60 epochs. For fair comparisons, all the baseline methods (i.e., AT, TRADES and MART) are trained for 80 epochs. **We have provided a detailed experiment setting that includes this information in Appendix F in the original submission.**
>
> > Q4. How does this affect the training performance?
>
> A4: Without the warm-up, the classifier is not properly learned initially, and thus may badly identify pixel regions that are important to the model's output in the early epochs. **Empirically, we add extra experiments to compare the performance of PART on CIFAR-10 with/without warm-up (see Table 4).**
>
> Table 4: Comparison of the performance of PART on CIFAR-10 with/without warm-up. The target model is ResNet-18. The number of warm-up epochs is 20. We report the averaged results and standard deviations of three runs. We show the most successful defense in **bold**.
> | Method | Natural     | PGD-20      | MMA         | AA          |
> |--------|-------------|-------------|-------------|-------------|
> | PART (with warm-up) | **83.42 ± 0.26** | **43.65 +- 0.16** | **41.98 ± 0.03** | **41.74 ± 0.04** |
> | PART (without warm-up) | 83.25 ± 0.17 | 43.27 +- 0.11 | 41.80 ± 0.08 | 41.36 ± 0.06 |
>
>
> > Q5. CAM technology appears to be a visualization technique primarily applicable to classification problems. How can the authors address concerns about the limited applicability of the PART framework in domains beyond classification? Are there plans to extend its use to other areas?
>
> A5: Thanks for your insightful question. You are right, CAM technology is initially designed for classification problems. However, one important thing we want to highlight is that **PART is a general idea rather than a specific method.** Specifically, PART is a general idea that we should count the fundamental discrepancies of pixel regions across images. **CAM technology, however, is only a tool to implement this idea.** If there is a tool that can be extended to other areas, we can integrate it into our framework. **The key is that the idea itself can be applied in other domains beyond classification, which is more important than how we implement it.** Similarly, most AT methods are designed to solve classification problems, but their ideas can be extended to other areas.
>
> > Q6. The clarity of results presentation is a concern. Could the authors provide more information about how standard deviations were calculated and specify the number of runs used in this calculation?
>
> A6: Thanks for your suggestion! We apologize that we missed stating the number of runs used in mean and standard deviation calculations. **We follow [1] and report the averaged results and standard deviations of 3 runs through the whole paper. We have specified this information in the updated version of our manuscript.** In our original submission, we did not report the standard deviations in Table 2 because of the limited page margin. **We have fixed this issue** by shrinking the font size to ‘footnotesize’, which also makes our tables more compact than before.
>
> [1]  Rahul Rade and Seyed-Mohsen Moosavi-Dezfooli. Reducing excessive margin to achieve a better accuracy vs. robustness trade-off. In *ICLR*, 2022.

---

> ### Author Response · Authors · 2023-11-19
> **Response to Reviewer h86N (part 3)**
>
> Thank you so much for your time and efforts spent on reviewing our paper! Your thorough main review and comments are very important to the improvement of our work! Please find our replies below.
>
> > Q7: Instead of comparing various CAM techniques, could the authors consider comparing the PART framework with a broader range of AT methods that utilize CAM technology, thereby offering a more comprehensive comparison of its effectiveness?
>
> A7: Thanks for your suggestion! To the best of our knowledge, we can find two adversarial defenses that incorporate CAM technologies. We have cited them in our related work (see Appendix C in the original submission for more details).
>
> Specifically, [1] proposes to use class activation features to remove adversarial noise. This method can be regarded as an adversarial purification method, which purifies adversarial examples towards natural examples. **Our method is fundamentally different from [1]** because that PART is an AT framework. Therefore, we cannot compare our method to [1] due to the different problem settings.
>
> On the other hand, [2] proposes a framework that uses GradCAM to rectify and preserve the visual attention area, which aims to improve the robustness against adversarial attacks by aligning the visual attention area between adversarial and original images. This method is an AT method and thus it is fair to compare our method to [2]. **However, we cannot find any source code of [2]. Therefore, it may take some time for us to realize their code. We will update the results once we finish the comparison.**
>
> [1] Dawei Zhou, Nannan Wang, Chunlei Peng, Xinbo Gao, Xiaoyu Wang, Jun Yu, and Tongliang Liu. Removing adversarial noise in class activation feature space. In *ICCV*, 2021.
>
> [2] Shangxi Wu, Jitao Sang, Kaiyuan Xu, Jiaming Zhang, and Jian Yu. Attention, please! adversarial defense via activation rectification and preservation. *ACM Trans. Multim. Comput. Commun. Appl*., 2023.
>
> > Q8. White-box attack defense results are discussed, but the impact of the number of attack iterations on the performance of CAM technology isn't addressed. Can the authors provide insights into how the number of attack iterations influences the effectiveness of CAM technology in adversarial defense?
>
> A8: **There might be a misunderstanding.** In Section 4.2 'Defending against adaptive attacks', we **have compared** our method to the baseline AT methods (i.e., AT, TRADES and MART) against adaptive PGD-20/40/60/80/100 (see Table 2 in the original submission for more details). From the experiment results, with the increase of attack iterations, the robustness of defense methods will decrease. This is because the possibility of finding worst-case examples will increase with more attack iterations. However, **the effectiveness of CAM technology itself is rarely influenced by attack iterations**, as we show that our method can consistently outperform baseline methods. Without losing generality, **we conduct an additional experiment** against normal PGD-10/40/60/80/100 (see Table 5). Again, we can obtain the same conclusions from the results. **We have included Table 5 in Appendix I in the updated version of our manuscript.**
>
> Table 5: Robustness (\%) of defense methods against PGD with different iterations on CIFAR-10. We report the averaged results and standard deviations of three runs. We show the most successful defense in **bold**.
> | Dataset   | Method | PGD-10              | PGD-40              | PGD-60              | PGD-80              | PGD-100             |
> |-----------|--------|---------------------|---------------------|---------------------|---------------------|---------------------|
> | CIFAR-10  | AT     | 44.83 ± 0.13        | 43.00 ± 0.10        | 42.83 ± 0.07        | 42.81 ± 0.03        | 42.81 ± 0.03        |
> |           | PART   | **45.20 ± 0.17**    | **43.20 ± 0.14**    | **43.09 ± 0.09**    | **43.08 ± 0.10**    | **42.93 ± 0.07**    |
> |           | TRADES | 48.81 ± 0.21        | 48.19 ± 0.13        | 48.16 ± 0.15        | 48.14 ± 0.08        | 48.08 ± 0.04        |
> |           | PART-T | **49.41 ± 0.11**    | **48.65 ± 0.10**    | **48.64 ± 0.13**    | **48.64 ± 0.04**    | **48.62 ± 0.03**    |
> |           | MART   | 49.98 ± 0.08        | 49.66 ± 0.16        | 49.66 ± 0.06        | 49.54 ± 0.03        | 49.47 ± 0.05        |
> |           | PART-M | **50.50 ± 0.19**    | **50.19 ± 0.15**    | **50.09 ± 0.04**    | **50.06 ± 0.05**    | **50.05 ± 0.02**    |

---

> ### Author Response · Authors · 2023-11-21
> **Reminder - Discussion Stage 1 closing soon - 21 November**
>
> Dear Reviewer h86N,
>
> We appreciate the time and effort that you have dedicated to reviewing our manuscript.  Just a quick reminder that discussion stage 1 is closing soon.
>
> Have our responses addressed your major concerns?
>
> If there is anything unclear, we will address it further. We look forward to your feedback.
>
> Best,
>
> Authors of Paper1033

---

> > ### Comment · Reviewer_h86N · 2023-11-22
> > **Further comments**
> >
> > Many thanks to the authors for their careful responses. I think the authors provide adequate explanations for most of the questions, so I will raise my rating. Whereas,
> >
> > 1. In the first table provided by the author, the experimental results of updating m every 10 epochs do not look satisfactory, and most of the results are not improved;
> >
> > 2. I would like to thank the authors for providing the results of the fifth table, but actually I was always interested that the authors could include some analysis of the effect of the number of attack iterations during training on the performance of the method in the subsequent versions.

---

> > > ### Author Response · Authors · 2023-11-22
> > > **Many thanks for your reply and increasing your score!**
> > >
> > > Dear Reviewer h86N,
> > >
> > > Many thanks for your valuable comments, and we are glad to hear that most of your concerns are well addressed. For your follow-up questions, please find our responses below.
> > >
> > > > Q1. In the first table provided by the author, the experimental results of updating m every 10 epochs do not look satisfactory, and most of the results are not improved.
> > >
> > > A1:  Updating m every 10 epochs will make our method much more efficient. At the same time, the natural accuracy can be further improved by a notable margin. In Table 1, with a marginal decrease of the robustness, **our method can always gain more natural accuracy (e.g., PART-M can gain 1.33% more natural accuracy if we update m every 10 epochs while the decrease in robustness is marginal).** Pointed by Reviewer QPEP, one of the main contributions of our paper is that **PART can improve the robustness-accuracy trade-off by notably increasing the natural accuracy.** After the discussion with Reviewer QPEP, we will emphasize this point more in the next version of our paper. We hope this could address your concerns with respect to the performance improvements.
> > >
> > > > Q2. I would like to thank the authors for providing the results of the fifth table, but actually I was always interested that the authors could include some analysis of the effect of the number of attack iterations during training on the performance of the method in the subsequent versions.
> > >
> > > A2: Thanks a lot for your suggestion! We will include the analysis in the next version of our paper once we finish the experiments!
> > >
> > > Best,
> > >
> > > Authors of Paper1033

---

> > > > ### Author Response · Authors · 2023-11-23
> > > > **Follow-up response**
> > > >
> > > > Dear Reviewer h86N,
> > > >
> > > > We take a close look at how the number of attack iterations during training would affect the final performance of CAM methods (see Table 1 below). Please find our reply to Q2 below:
> > > >
> > > > If we increase the attack iterations during training, the model will become more robust as the model learns more worst-case examples during training. At the same time, the natural accuracy has a marginal decrease, which is probably due to the inherent trade-off between natural accuracy and robustness [1]. Overall, we can obtain the same conclusions as Q8 we discussed before: the performance of our method is stable and CAM method is rarely affected by the attack iterations.
> > > >
> > > > Table 1: Robustness (\%) and Accuracy (\%) of PART against PGD with different iterations during training on CIFAR-10. The target model is ResNet-18. We report the averaged results and standard deviations of three runs.
> > > >
> > > > |Dataset|Method|Natural|PGD-20|MMA| AA|
> > > > |-----|----|---|---|---|---|
> > > > |CIFAR-10| PART (PGD-10)| 83.42 ± 0.26| 43.65 ± 0.16| 41.98 ± 0.03| 41.74 ± 0.04|
> > > > ||PART (PGD-20)|83.44 ± 0.19|43.64 ± 0.13|42.02 ± 0.13|41.82 ± 0.08|
> > > > ||PART (PGD-40)|83.36 ± 0.21|43.82 ± 0.08|42.09 ± 0.07|41.86 ± 0.11|
> > > > ||PART (PGD-60)|83.30 ± 0.15|44.02 ± 0.12|42.18 ± 0.05| 41.91 ± 0.09|
> > > >
> > > > [1] Dimitris Tsipras, Shibani Santurkar, Logan Engstrom, Alexander Turner, and Aleksander Madry. Robustness may be at odds with accuracy. In *ICLR*, 2019.
> > > >
> > > > We have included the analysis in the updated version of our manuscript (hilighted in red). Please check Appendix I for more details.
> > > >
> > > > If there is anything unclear, we will address it further. If we have addressed all the concerns, please kindly re-evaluate our paper.
> > > >
> > > > Best,
> > > >
> > > > Authors of Paper1033

---

### Official Review · Reviewer_tC3M · 2023-10-26

**Soundness:** 3 good
**Presentation:** 3 good
**Contribution:** 3 good
**Rating:** 8
**Confidence:** 4

**Summary:**

This submission proposes pixel-wise reweighting for adversarial training. The central observation presented in this work is that not all pixels of the adversarial perturbation contribute equally to the accuracy of the model. The authors propose a new framework for adversarial training called pixel-reweighed adversarial training (PART) which uses class activation mapping to identify important pixel regions. Authors evaluate their adversarial training framework on the CIFAR-10, SVHN, and Tiny-ImageNet datasets using a ResNet
and a WideResNet model.

**Strengths:**

- The paper presents theoretical results on a toy model which help understanding the method.
- The empirical evaluation is solid and the proposed PART method is compared against vanilla adversarial training, TRADES, and MART.
- The paper reads mostly very well and is very understandable. I only found a few typos (see below).

**Weaknesses:**

- The idea is not completely novel. Adversarial attacks in combination with class activation mappings have for example been discussed in [1]. However, the authors use it for robustifying their models which is in my opinion sufficiently different. Nonetheless, authors should include a citation of that work.
- The empirical evaluation can be extended by using different adversarial attacks, e.g., Carlini-Wagner attack or AutoAttack.
- The literature review seems somewhat short. I suggest authors spend more time looking for relevant related works.
- Performence of the PART method is somewhat underwhelming. The improvement is only incremental (usually only in the range of ~1%).
- Section 3.1 “AE generation process.” is tough to read. Authors should work on the presentation of that section. Maybe a small table on the side would help to introduce the notation.
- Figure 4: Authors should mention what is indicated by the shaded areas.

Overall, the ideas in this paper are not ground-breaking, but the solid theoretical and empirical analysis justify its publication in ICLR, which is why I recommend to accept this submission.

Minor details:
- Missing whitespace “Table 2: Robustness(%) of…”
- Eq. 12-15 “subject to” should not be typeset in math mode
- Lemma 1: “(i).” unusual period

References

[1] Dong, Xiaoyi, et al. "Robust superpixel-guided attentional adversarial attack." Proceedings of the IEEE/CVF Conference on Computer Vision and Pattern Recognition. 2020.

**Questions:**

- Figure 3: What should the lock symbol next to the first CNN tell me?
- Figure 3: In what sense are the activation maps of that CNN “global”?

---

> ### Author Response · Authors · 2023-11-19
> **Response to Reviewer tC3M (part 1)**
>
> Thank you so much for your positive comments and suggestions! It is our pleasure that our theoretical and empirical analysis can be recognized. Your thorough main review and comments are very important to the improvement of our work! Please find our replies below.
>
> > Q1. The idea is not completely novel. Adversarial attacks in combination with class activation mappings have for example been discussed in [1]. However, the authors use it for robustifying their models which is in my opinion sufficiently different. Nonetheless, authors should include a citation of that work.
> > [1] Dong, Xiaoyi, et al. "Robust superpixel-guided attentional adversarial attack." Proceedings of the IEEE/CVF Conference on Computer Vision and Pattern Recognition. 2020.
>
> A1: Thanks for your suggestion! **We have cited this paper in our related work in Appendix C in the updated version of our manuscript (highlighted in blue).** Here is the added content:
>
> **Adversarial attacks with class activation mapping.** Dong et al. (2020) proposes an attack method that leverages superpixel segmentation and class activation mapping to focus on regions of an image that are most influential in classification decisions. It highlights the importance of considering perceptual features and classification-relevant regions in crafting effective AEs.
>
> > Q2: The empirical evaluation can be extended by using different adversarial attacks, e.g., Carlini-Wagner attack or AutoAttack.
>
> A2: Thanks for your suggestion! **However, there might be a misunderstanding. For general attacks, we have evaluated our methods and baselines using AutoAttack (see Table 1 in the original submission).** For adaptive attacks, we conduct an additional experiment to test the robustness of defense methods against adaptive MMA (see Table 1 below). The choice of MMA over AA for adaptive attacks is due to AA's time-consuming nature as an ensemble of multiple attacks. Incorporating the CAM method into AA would further slow the process. MMA, in contrast, offers greater time efficiency and comparable performance to AA. **We have added Table 1 below in Appendix J, please refer to the updated version of our manuscript.**
>
> Table 1: Robustness (\%) of defense methods against adaptive MMA on CIFAR-10. We report the averaged results and standard deviations of three runs. We show the most successful defense in **bold**.
>
> | Dataset   | Method | MMA-20              | MMA-40              | MMA-60              | MMA-80              | MMA-100             |
> |-----------|--------|---------------------|---------------------|---------------------|---------------------|---------------------|
> | CIFAR-10  | AT     | 35.36 ± 0.10        | 35.02 ± 0.05        | 34.93 ± 0.09        | 34.86 ± 0.06        | 34.85 ± 0.07        |
> |           | PART   | **35.67 ± 0.07**    | **35.35 ± 0.11**    | **35.29 ± 0.13**    | **35.29 ± 0.09**    | **35.17 ± 0.05**    |
> |           | TRADES | 40.14 ± 0.08        | 39.89 ± 0.12        | 39.93 ± 0.05        | 39.87 ± 0.08        | 39.82 ± 0.03        |
> |           | PART-T | **40.78 ± 0.13**    | **40.57 ± 0.11**    | **40.51 ± 0.08**    | **40.49 ± 0.05**    | **40.48 ± 0.02**    |
> |           | MART   | 39.14 ± 0.06        | 38.79 ± 0.13        | 38.80 ± 0.10        | 38.79 ± 0.05        | 38.74 ± 0.08        |
> |           | PART-M | **40.56 ± 0.11**    | **40.26 ± 0.07**    | **40.23 ± 0.12**    | **40.21 ± 0.08**    | **40.20 ± 0.07**    |
>
> > Q3. The literature review seems somewhat short. I suggest authors spend more time looking for relevant related works.
>
> A3: Due to the limited space, the literature review in the main body of the paper is short. **However, we have provided a detailed literature review in Appendix C,** which covers the literature review of different AT methods, CAM methods, adversarial defenses with CAM and adversarial attacks with CAM. **We have also included the discussion regarding the literature suggested in your Q1.** Please check the full literature review in the updated version of our manuscript.

---

> ### Author Response · Authors · 2023-11-19
> **Response to Reviewer tC3M (part 2)**
>
> Thank you so much for your positive comments and suggestions! It is our pleasure that our theoretical and empirical analysis can be recognized. Your thorough main review and comments are very important to the improvement of our work! Please find our replies below.
>
> > Q4. Performence of the PART method is somewhat underwhelming. The improvement is only incremental (usually only in the range of ~1%).
>
> A4: It is reasonable that our method gains marginal improvement in robustness, as PART decreases the attack strength for unimportant pixel regions. **What we want to highlight is that PART can improve the robustness-accuracy trade-off by notably increasing the natural accuracy.** Despite a marginal reduction in robustness by 0.04\% on PGD-20, PART gains more on natural accuracy (e.g., 2.08\% on SVHN and 1.36\% on Tiny-ImageNet). In most cases, PART can improve natural accuracy and robustness simultaneously. Compared to TRADES and MART, our method can still boost natural accuracy (e.g., 1.20\% on CIFAR-10, 2.64\% on SVHN and 1.26\% on Tiny-ImageNet for PART-T, and 1.85\% on CIFAR-10, 1.66\% on SVHN and 1.07\% on Tiny-ImageNet) with at most a 0.10\% drop in robustness. It is widely acknowledged that AT will hurt natural accuracy compared to standard training. **Our work can bridge this gap without compromising robustness.** As demonstrated in [1], advancements in the field of AT have been incremental over the past several years. As a result, making every slight improvement in this area is noteworthy.
>
> [1]  Rahul Rade and Seyed-Mohsen Moosavi-Dezfooli. Reducing excessive margin to achieve a better accuracy vs. robustness trade-off. In *ICLR*, 2022.
>
> > Q5. Section 3.1 “AE generation process.” is tough to read. Authors should work on the presentation of that section. Maybe a small table on the side would help to introduce the notation.
>
> A5: Thanks for your suggestion! **We have uploaded a reference list of the notations used in Section 3.1 in Appendix D. Please check the updated version of our manuscript.** For your convienience, we also show the notation table below.
>
> | Symbol | Description |
> | ------ | ----------- |
> | $\ell$ | A loss function |
> | $f$ | A model |
> | $\mathbf{x}$ | A natural image |
> | $y$ | The true label of $\mathbf{x}$ |
> | $d$ | The data dimension |
> | $\mathbf{\Delta}$ | The adversarial perturbation added to $\mathbf{x}$ |
> | $\mathbf{\Delta}^*$ | The optimal solution of  $\mathbf{\Delta}$ |
> | $\|\cdot\|_{\infty}$ | The $\ell_{\infty}$-norm |
> | $\epsilon$ | The maximum allowed perturbation budget for important pixels |
> | $\epsilon^{\rm low}$ | The maximum allowed perturbation budget for unimportant pixels |
> | $\mathcal{I}^{\rm high}$ | Indexes of important pixels |
> | $\mathcal{I}^{\rm low}$ | Indexes of unimportant pixels |
> | $\bf{v}$ | A function to transform a set to a vector |
> | $\\{\delta_i\\}_{i \in \mathcal{I}^{\rm high}}$ | A set consisting of important pixels in $\mathbf{\Delta}$, i.e., $\mathbf{\Delta}^{\rm high}$ |
> | $ \\{\delta_i\\}_{i \in \mathcal{I}^{\rm low}}$ | A set consisting of unimportant pixels in $\mathbf{\Delta}$, i.e., $\mathbf{\Delta}^{\rm low}$ |
> | $\|\mathcal{I}^{\rm high}\|$ | The dimension of important pixel regions, i.e., $d^{\rm high}$ |
> | $\|\mathcal{I}^{\rm low}\|$ | The dimension of unimportant pixel regions, i.e., $d^{\rm low}$ |
>
> > Q6. Figure 4: Authors should mention what is indicated by the shaded areas.
>
> A6: Thanks for your suggestion! We apologize that we missed stating the meaning of the shaded areas. **The shaded areas represent the standard deviation. We have included this information in the updated version of our manuscript.**
>
> > Q7. Minor details (e.g., missing whitespace “Table 2: Robustness(%) of…” , Eq. 12-15 “subject to” should not be typeset in math mode and Lemma 1: “(i).” unusual period).
>
> A7: Thank you for pointing out these minor issues in our paper! **We have fix these issues in the updated version of our manuscript.**
>
> > Q8. Figure 3: What should the lock symbol next to the first CNN tell me?
>
> A8: The lock symbol means **the parameters of the model are fixed**. This is because during the generation of AE, the parameters of the model are unchanged. **We have explained the meaning of the lock symbol in the legend of the figure in our updated version of manuscript.**
>
> > Q9. Figure 3: In what sense are the activation maps of that CNN “global” ?
>
> A9: We apologize for the misuse the word 'global'. The feature maps generated by CAM methods are actually a form of visual explanation, showing which parts of the input image the network focuses on when making decisions. These feature maps usually concentrate more on local regions (i.e., the areas of the image most crucial for the classification decision) rather than the entire image. Therefore, referring to the feature maps extracted by CAM methods as 'global feature maps' might not be accurate. **To avoid confusion, we have removed the word 'global' in the updated version of our manuscript.**

---

> > ### Comment · Reviewer_tC3M · 2023-11-21
> >
> > Thank you for the clarification and the additional results. My concerns were adequately addressed and I have raised my score to 8.

---

> > > ### Author Response · Authors · 2023-11-21
> > > **Many thanks for your reply and increasing your score to 8!**
> > >
> > > Dear Reviewer tC3M,
> > >
> > > Many thanks for your valuable comments again, and we are glad to hear that your concerns are well addressed.
> > >
> > > Best regards,
> > >
> > > Authors of Paper1033

---

> ### Author Response · Authors · 2023-11-21
> **Reminder - Discussion Stage 1 closing soon - 21 November**
>
> Dear Reviewer tC3M,
>
> We appreciate the time and effort that you have dedicated to reviewing our manuscript.  Just a quick reminder that discussion stage 1 is closing soon.
>
> Have our responses addressed your major concerns?
>
> If there is anything unclear, we will address it further. We look forward to your feedback.
>
> Best,
>
> Authors of Paper1033

---

### Official Review · Reviewer_u6mu · 2023-10-31

**Soundness:** 3 good
**Presentation:** 3 good
**Contribution:** 2 fair
**Rating:** 6
**Confidence:** 3

**Summary:**

The paper presents Pixel-reweighted Adversarial Training (PART), a adjusted adversarial training framework designed to enhance model robustness against adversarial attacks. PART introduces a dynamic perturbation allocation strategy, redistributing the perturbation across pixels according to their influence on model output. This is a departure from traditional AT methods, which apply a uniform noise across all pixels. It tested on the common benchmarks to proof that such reweighted perturbation enhances model performance by allowing the model to focus on more critical areas of the image that significantly impact model decisions.

**Strengths:**

This paper takes a step further on the adversarial training. The concept of pixel influence on robustness and accuracy is well-motivated, grounded on the premise that not all parts of an image equally contribute to the decision-making process of a neural network. Overall, the paper is easy to follow.

The authors conducted experiments on common benchmarks, including CIFAR-10, SVHN and Tiny-ImageNet. Extensive results show the proposed PART generally outperforms standard AT.

**Weaknesses:**

I am not surprised by the proposed method that introducing CAM to direction adversarial training to the semantic meaningful regions.

Also, I am unsure if the proposed method can be scaled up --- due to (1) CAM may not be scalable which means it may lose the ability to identify the semantic meaningful regions; (2) the author currently did not analyze the computation cost and training time cost of the proposed PART compared to AT and standard training.

# Post-rebuttal

I raised my score due to author provide more detailed comparisons. However, I am unsure if the proposed framework can be scale-up or not. Hope AC can examine this point.

**Questions:**

I would be curious if the proposed method applied into larger dataset which also obtains improvements.

---

> ### Author Response · Authors · 2023-11-19
> **Response to Reviewer u6mu**
>
> Thank you so much for your valuable comments and questions! Please find our replies below.
>
> > Q1. CAM may not be scalable which means it may lose the ability to identify the semantic meaningful regions.
>
> A1: **Based on our experiment results on Tiny-ImageNet, we believe CAM can be scaled to large datasets.** The scalability of CAM methods to large datasets primarily depends on the scalability of the model it integrates with. CAM itself is a visualization technique used to highlight the regions of an image that CNNs focus on when making decisions. If the network itself can effectively process large datasets (e.g., having sufficient learning capacity and appropriate computational efficiency), then CAM can also be applied to these datasets. Due to the high computational cost of AT, conducting experiments on larger datasets such as ImageNet would require significant resources. Typically, **Tiny-ImageNet, a scaled-down version of ImageNet, is already a relatively large dataset used in the field of AT.** When applying our method to a very large dataset such as ImageNet, we can **scale down** the size of images to fit the capacity of our network, and thus CAM methods will not lose the ability to identify the semantic meaningful regions.
>
> > Q2. The author currently did not analyze the computation cost and training time cost of the proposed PART compared to AT and standard training.
>
> A2: Incorporating CAM technologies will naturally lead to an increase in training speed. How to effectively integrate these technologies into the training process presents an intriguing challenge. If we generate the mask m for each epoch, the computational cost will be considerably high. **To address this problem, we update the mask m every 10 epochs.** We show that the performance of our method remains competitive (see Table 1) given the mask is updated for every 10 epochs. Regarding memory consumption, the majority of the memory is allocated for storing checkpoints, with only a small portion attributed to CAM technology. In general, the additional costs (i.e., training speed and memory consumption) are affordable (see Table 2 and 3). **This information has been included in the updated version of our manuscript.** For more details, please refer to Appendix M in the updated version.
>
> Table 1: Robustness (\%) of defense methods on CIFAR-10. The target model is ResNet-18. We report the averaged results and standard deviations of three runs. We show the most successful defense in **bold**.
> | Method | Natural | PGD-20| MMA  | AA |
> |--------|-------------|-------------|-------------|-------------|
> | AT     | 82.58 ± 0.14| **43.69 ± 0.28**| 41.80 ± 0.10| 41.63 ± 0.22|
> | PART (update m for every epoch) | 83.42 ± 0.26 | 43.65 +- 0.16 | **41.98 ± 0.03** | **41.74 ± 0.04** |
> | PART (update m for every 10 epochs) | **83.77 ± 0.15** | 43.36 +- 0.21 | 41.83 ± 0.07 | 41.41 ± 0.14 |
> | TRADES | 78.16 ± 0.15| 48.28 ± 0.05| 45.00 ± 0.08| 45.05 ± 0.12|
> | PART-T (update m for every epoch) | 79.36 ± 0.31 | **48.90 ± 0.14**|**45.90 ± 0.07**| **45.97 ± 0.06** |\
> | PART-T (update m for every 10 epochs) | **80.13 ± 0.16** | 48.72 ± 0.11|45.59 ± 0.09|45.60 ± 0.04|
> | MART   | 76.82 ± 0.28 | 49.86 ± 0.32  | 45.42 ± 0.04| 45.10 ± 0.06 |
> | PART-M (update m for every epoch) |78.67 ± 0.10| **50.26 ± 0.17**|**45.53 ± 0.05** | **45.19 ± 0.04**|
> | PART-M (update m for every 10 epochs) |**80.00 ± 0.15**| 49.71 ± 0.12|45.14 ± 0.10| 44.61 ± 0.24|
>
> Table 2: Training speed (hours:minutes:seconds) of defense methods on CIFAR-10. The target model is ResNet-18.
> |GPU |Method  | Training Speed | Difference |
> |-----------------|--------|----------------|------------|
> | 1 \* NVIDIA A100| SAT| 02:14:37| \ |
> | | PART| 02:43:45| 00:29:08   |
> | | TRADES | 02:44:19| \ |
> | | PART-T | 03:15:06| 00:30:47   |
> | | MART   | 02:09:23| \ |
> | | PART-M | 02:39:37 | 00:30:14   |
>
> Table 3: Memory consumption (MB) of defense methods on CIFAR-10. The target model is ResNet-18.
> | Method   | Memory Consumption | Difference |
> |-----|-----|-----|
> | SAT | 5530MB | \ |
> | PART| 5877MB| 347MB|
> | TRADES| 5369MB| \ |
> | PART-T | 5688MB | 319MB |
> | MART | 5553MB  | \ |
> | PART-M| 5894MB  | 341MB |
>
>
> > Q3. I would be curious if the proposed method applied into larger dataset which also obtains improvements.
>
> A3: On Tiny-ImageNet, our method can consistently improve the performance compared to baseline methods. If the 'larger' here refers to the number of categories, Tiny-ImageNet with 200 categories can be considered a relatively large dataset. If the 'larger' here refers to the image dimensions, we can scale down the image size to fit the model capacity. As mentioned in Q1, the scalability of CAM methods to large datasets primarily depends on the scalability of the model it integrates with. In terms of the computation cost and memory consumption, we have provided a detailed comparison in Q2. In general, **based on our experiment results on Tiny-ImageNet, we believe CAM can be scaled to large datasets.**

---

> ### Author Response · Authors · 2023-11-21
> **Reminder - Discussion Stage 1 closing soon - 21 November**
>
> Dear Reviewer u6mu,
>
> We appreciate the time and effort that you have dedicated to reviewing our manuscript.  Just a quick reminder that discussion stage 1 is closing soon.
>
> Have our responses addressed your major concerns?
>
> If there is anything unclear, we will address it further. We look forward to your feedback.
>
> Best,
>
> Authors of Paper1033

---

> ### Author Response · Authors · 2023-11-22
> **Reminder - Discussion Stage 1 closing soon - 22 November**
>
> Dear Reviewer u6mu,
>
> We appreciate the time and effort that you have dedicated to reviewing our manuscript. Just a quick reminder that discussion stage 1 is closing soon.
>
> Have our responses addressed your major concerns?
>
> If there is anything unclear, we will address it further.
>
> Your feedback is very important to our work. We look forward to your feedback.
>
> Best,
>
> Authors of Paper1033

---

> ### Author Response · Authors · 2023-11-23
> **Reminder - Discussion Stage 1 closing soon - 23 November**
>
> Dear Reviewer u6mu,
>
> We appreciate the time and effort that you have dedicated to reviewing our manuscript. Just a quick reminder that discussion stage 1 is closing soon.
>
> Have our responses addressed your major concerns?
>
> If there is anything unclear, we will address it further.
>
> Your feedback is very important to our work. We look forward to your feedback.
>
> Best,
>
> Authors of Paper1033

---

> > ### Comment · Reviewer_u6mu · 2023-11-23
> > **Reviewer Response**
> >
> > I appreciate the efforts made by the authors. I believe adding the detailed time and memory cost can help readers understand the performance of the proposed method.
> >
> > I raised my score, while I'd like to confirm that your wordings cannot solve my concerns about if the proposed method can be scaled up or not. I will mention my concern to AC in this aspect.

---

> > > ### Author Response · Authors · 2023-11-23
> > > **Many thanks for your reply and increasing your score to 6!**
> > >
> > > Dear Reviewer u6mu,
> > >
> > > Many thanks for your support!
> > >
> > > As for whether our method can be scaled up or not, we find that it might be helpful to analyze if the algorithm running complexity will linearly increase when linearly increasing the number of samples or data dimensions (not just by conducting experiments). We are now working on that and hope to get back to you with our analysis soon.
> > >
> > > Best regards,
> > >
> > > Authors of Paper1033

---

> > > ### Author Response · Authors · 2023-11-23
> > > **Follow-up Response**
> > >
> > > Dear Reviewer u6mu,
> > >
> > > Thanks again for your valuable comments! Please find our follow-up response below:
> > >
> > > > Q: I raised my score, while I'd like to confirm that your wordings cannot solve my concerns about if the proposed method can be scaled up or not.
> > >
> > > A: The generation of class activation mapping **mainly involves global average pooling**, which has a complexity of `O(H_last × W_last × C_last)`, where `H_last`, `W_last`, and `C_last` are the height, width, and number of channels in the last convolutional layer. The subsequent generation of the heatmap involves **a weighted combination of these pooled values with the feature maps**, which also has a complexity **proportional to** the size of the feature map. When data linearly increases, the complexity of Grad-CAM is mainly influenced by two factors: **the size of the input data** and **the structure of the CNN**. Considering the case of a single input sample, **the increase in input data primarily refers to the increase in the dimensions of the input image.** Suppose the width and height of the image both increase linearly, for instance, doubling in size, then the total number of pixels in the input image will quadruple (since the area is the product of width and height). This will **linearly** increase the computational complexity of the convolutional layers, as each convolution operation will need to process more pixels. However, since convolution is a local operation, **this increase is generally manageable.** Therefore, for a single sample, the computational complexity of Grad-CAM in relation to the growth of input data is **approximately linear**. When considering multiple samples, i.e., **the linear increase in the size of the dataset, Grad-CAM will need to process more samples.** In this case, the total computational complexity will be **linearly related to the number of samples**.
> > >
> > > We hope this could address your concern. If there is anything unclear, we will address it further.
> > >
> > > Best,
> > >
> > > Authors of Paper1033

---

### Official Review · Reviewer_QPEP · 2023-11-07

**Soundness:** 2 fair
**Presentation:** 3 good
**Contribution:** 2 fair
**Rating:** 6
**Confidence:** 3

**Summary:**

This paper introduces a new framework of pixel-based re-weighting to gauge a more robust way of adversarial training. Authors begin with a proof of concept example showing how not all part of an image are equally informative, and then proceed to create an automated pipeline for adversarial training that can generalize and extend to multiple images based on gradient-based methods that show what parts of the image activate a certain class the most (parts that will later be weight more aggressively for the attack). Authors finish the paper with additional quantitative plots. I wish authors would have shown at the end how their adversarial images for networks trained on PART look like.

**Strengths:**

* The paper introduces a way to improve adversarial robustness through part-based re-weighting given the interesting parts of information in an image.
* Authors propose a modular framework that can be used for future adversarial training pipelines.
* Authors show how PART is better than many other adversarial training pipelines but the increase is very incremental. Should the paper be accepted because of this last point? I am not entirely convinced.

**Weaknesses:**

* The paper says towards the end that this framework in more aligned to human perception. I don't think this is true from what has been shown in the paper. I would have liked to see qualitative samples and attack comparing how PGD performs on the same network trained differently (without AT, and with AT either classical or PART based), and from there run a psychophysical experiment with human observers to see if indeed they are fooled more by the PART-based model. While running the psychophysical experiments may not be possible, even adding the resulting adversarial images from networks trained with PART would be a great addition to the paper.

* I'd really recommend plotting the adversarial images for networks trained on PART similar to how this was done in Santurkar et al. (NeurIPS 2019), Berrios & Deza (ArXiv 2022) and Gaziv et al. (NeurIPS 2023).

**Questions:**

I think this paper is interesting but I am on fence of the contribution. Are all pixels equally important in an image? My gut feeling says that the answer is No, and perhaps it's a bit of a tautology (this seems quite obvious). Would it not be possible that performing adversarial training end-to-end with PGD-based type image diets automatically help a neural network find these critical parts in the image from which to then perturb the image at training? All-in-all, my question is: is PART really useful when there are many other adversarial training regimes that go beyond FGSM and that implicitly incorporate the image structure in the adversarial optimization?

I am not an expert in the adversarial robustness literature, so I am curious to hear what other reviewers say about this proposed training framework. I am willing to change my mind depending on the rebuttal and on knowing what the other reviews have to say about this paper.

Perhaps another interesting question that I would have liked authors answer is. How would PART work with training on other image distributions such as Textures or Scenes? Would PART be equally useful or will it only apply to objects?

---

> ### Author Response · Authors · 2023-11-19
> **Response to Reviewer QPEP (part 1)**
>
> Thank you so much for your valuable suggestions and intetersting questions! Please find our replies below.
>
> > Q1. Authors show how PART is better than many other adversarial training pipelines but the increase is very incremental. Should the paper be accepted because of this last point? I am not entirely convinced.
>
> It is reasonable that our method gains marginal improvement in robustness, as PART decreases the attack strength for unimportant pixel regions. **What we want to highlight is that PART can improve the robustness-accuracy trade-off by notably increasing the natural accuracy.** Despite a marginal reduction in robustness by 0.04\% on PGD-20, PART gains more on natural accuracy (e.g., 2.08\% on SVHN and 1.36\% on Tiny-ImageNet). In most cases, PART can improve natural accuracy and robustness simultaneously. Compared to TRADES and MART, our method can still boost natural accuracy (e.g., 1.20\% on CIFAR-10, 2.64\% on SVHN and 1.26\% on Tiny-ImageNet for PART-T, and 1.85\% on CIFAR-10, 1.66\% on SVHN and 1.07\% on Tiny-ImageNet) with at most a 0.10\% drop in robustness. It is widely acknowledged that AT will hurt natural accuracy compared to standard training. **Our work can bridge this gap without compromising robustness.** As demonstrated in [1], advancements in the field of AT have been incremental over the past several years. As a result, making every slight improvement in this area is noteworthy.
>
> [1]  Rahul Rade and Seyed-Mohsen Moosavi-Dezfooli. Reducing excessive margin to achieve a better accuracy vs. robustness trade-off. In *ICLR*, 2022.
>
> > Q2. The paper says towards the end that this framework in more aligned to human perception. I don't think this is true from what has been shown in the paper. I would have liked to see qualitative samples and attack comparing how PGD performs on the same network trained differently (without AT, and with AT either classical or PART based), and from there run a psychophysical experiment with human observers to see if indeed they are fooled more by the PART-based model. While running the psychophysical experiments may not be possible, even adding the resulting adversarial images from networks trained with PART would be a great addition to the paper. I'd really recommend plotting the adversarial images for networks trained on PART similar to how this was done in Santurkar et al. (NeurIPS 2019), Berrios & Deza (ArXiv 2022) and Gaziv et al. (NeurIPS 2023).
>
> A2: Thanks for your great recommendation and thanks for bringing these insightful papers to us, **but there might be a misunderstanding.** One thing we would like to point out is that 'more aligned to human perception' **does not mean** that AEs generated by PART-based methods can fool more human observers. As long as the adversarial noise itself is bounded by a small $\epsilon$ (i.e., imperceptible to human eyes), every adversarial attack can fool human observers. In our paper, **'the alignment to human perception' refers to how well the model's attention aligns with the object (i.e., semantic meaning) in an image.** We find that the previously stated sentence may cause unnecessary confusion. Thus, we have changed **'human-aligned information'** into **'semantic information'** in the updated version of our manuscript.
>
> We agree with your comment that adding the resulting adversarial images from networks trained with PART would be a great addition to the paper. In our original submission, we **have provided** examples of the high-contribution pixel regions learnt by the same network trained differently (e.g., without AT, with AT and with PART-based) in Appendix N. In Figure 6, it is clear that the network trained without AT tends to focus on the background of the image, and the network trained with PART is **more precisely aligned with the object** than the network trained with AT.
>
> To further support our statements, **we conduct additional experiments** to visualize how the high-contribution pixel regions change every 10 epochs from the 30th epoch to the 60th epoch (see Figures 7 and 8 in the updated version of our manuscript). With the increase of epochs, PART-based methods are more precisely aligned to the object (i.e., semantic meaning) in an image compared to baseline AT methods.

---

> ### Author Response · Authors · 2023-11-19
> **Response to Reviewer QPEP (part 2)**
>
> Thank you so much for your valuable suggestions and intetersting questions! Please find our replies below.
>
> > Q3. I think this paper is interesting but I am on fence of the contribution. Are all pixels equally important in an image? My gut feeling says that the answer is No, and perhaps it's a bit of a tautology (this seems quite obvious).
>
> A3: Thanks for thinking our paper is interesting. However, in our humble opinions, it is not very fair to say our contribution is limited just because this paper has a straightforward intuition. Instead, we treat this as a strength of our paper and we would like to elaborate on this point. While the intuition behind our paper might appear straightforward, its simplicity should not be mistaken for a lack of depth or contribution. We have validated this intuitive idea from both empirical and theoretical standpoints. Our work not only demonstrates the practical applicability of our intuition but also provides theoretical insights that reinforces its validity. We want to emphsize that **simple intuitions can also lead to significant advancements, as long as they are well investigated and validated.**
>
>
> > Q4. Would it not be possible that performing adversarial training end-to-end with PGD-based type image diets automatically help a neural network find these critical parts in the image from which to then perturb the image at training? All-in-all, my question is: is PART really useful when there are many other adversarial training regimes that go beyond FGSM and that implicitly incorporate the image structure in the adversarial optimization?
>
> A4: It is possible. PGD-based attacks aim to find the minimum perturbation that can fool the neural network. This process often highlights the most sensitive and critical features of the image that the network relies on for making decisions. **However, the problem is that we find these critical features learnt by PGD-based AT are not sufficient.** This is also one of the key contributions of our paper. We find that without explicit guidance, it is hard for AT methods to sufficiently mine robust features. **The advantage of PART, on the other hand, is the ability to provide external guidance to help models better extract features that are beneficial to robust classification.**
>
> > Q5. Perhaps another interesting question that I would have liked authors answer is. How would PART work with training on other image distributions such as Textures or Scenes? Would PART be equally useful or will it only apply to objects?
>
> A5: Thanks for your insightful question! **The applicability of PART primarily depends on the applicability of the model it integrates with.** PART is a general idea rather than a specific method and CAM technology is only a tool to implement this idea. As long as the network itself can effectively work with Textures or Scenes (i.e., can extract meaningful classification patterns), then PART is able to obtain high-contribution pixel regions which means PART will still be useful on these image distributions.

---

> ### Author Response · Authors · 2023-11-21
> **Reminder - Discussion Stage 1 closing soon - 21 November**
>
> Dear Reviewer QPEP,
>
> We appreciate the time and effort that you have dedicated to reviewing our manuscript.  Just a quick reminder that discussion stage 1 is closing soon.
>
> Have our responses addressed your major concerns?
>
> If there is anything unclear, we will address it further. We look forward to your feedback.
>
> Best,
>
> Authors of Paper1033

---

> > ### Comment · Reviewer_QPEP · 2023-11-21
> > **Comments addressed, Increasing my Score**
> >
> > Dear Authors,
> >
> > My concerns have been addressed and I've raised my score from 5 to 6. The main point for me that has helped me raise my score is that the term "perceptual alignment" has been corrected to "semantic information", and also that I think the sentence above makes me interpret better the results:
> >
> > > **What we want to highlight is that PART can improve the robustness-accuracy trade-off by notably increasing the natural accuracy**
> >
> > I think this should be highlighted more in the Abstract or Introduction, as I would have not have guessed that was one of the main contributions from reading the paper and looking at the figures on a first glance.
> >
> > I think this paper would make a good poster at the conference, and I have enjoyed reading the other reviews and rebuttals.

---

> > > ### Author Response · Authors · 2023-11-21
> > > **Many thanks for your reply and increasing your score to 6!**
> > >
> > > Dear Reviewer QPEP,
> > >
> > > Many thanks for your valuable comments again, and we are glad to hear that your concerns are well addressed.
> > >
> > > We totally agree that the sentence you highlighted is very important. We will emphasize this point more in the next version of our paper.
> > >
> > > Best regards,
> > >
> > > Authors of Paper1033

---

### Meta-Review · Area_Chair_xEpt · 2023-12-13

**Metareview:**

This paper proposes Pixel-reweighted adversarial training (PART), an improvement to Adversarial Training by reweighting the perturbation-norm budget for each pixel in the input image based on a class activation map (CAM). Because CAM requires a target class, PART only works for targetted attacks (however, this limitation in the paper and the paper is missing a Limitations section).

The work shows extensive results that PART outperforms state-of-the-art baselines consistently by a small margin (mostly ~1% in success rates on these datasets) on 3 small-scale datasets (CIFAR-10, SVHN, TinyImageNet) and 2 architectures (ResNet-18 and WideResNet-34-10). Reviewers (`QPEP` and `tC3M`) are concerned that the contribution of the work might be too incremental or hard to scale up to larger images (reviewer `u6mu`).

Overall, I agree with the reviewers that the empirical results are extensive. But I also agree that the contribution is marginal. The areas for improvement are:
- PART should be tested on ImageNet (similar to state-of-the-art attack methods) instead of TinyImageNet
- The authors might want to demonstrate how qualitatively the perturbation appears less perceptible (compared to AT and baselines). This is also raised by the reviewer `QPEP`. This might an extra advantage of PART.
- There are other very relevant works that also regularize the attention heatmap to improve the robustness of models e.g. [1]. The idea is also to train the model to regularize the attention map of ViTs to focus on classification objects, hence improving the robustness of models.
[2] shows that regularizing the gradient-based attribution maps also improves model robustness. The paper is missing these two works in their literature review.
- This idea may work as well for ViT!

[1] Optimizing Relevance Maps of Vision Transformers Improves Robustness. NeurIPS 2022

[2] Improving the Adversarial Robustness and Interpretability of Deep Neural Networks by Regularizing Their Input Gradients. AAAI 2017.

Overall, I find this work to be on the borderline and encourage the authors to keep working to make the contribution of the methods stronger for future revisions.

**Justification For Why Not Higher Score:**

Contribution is marginal since all 3 datasets are small-scale, and no qualitative results of the adversarial images are shown.

**Justification For Why Not Lower Score:**

N/A

---

### Decision · Program_Chairs · 2024-01-16

Reject